# Stress-induced vagal activity influences anxiety-relevant prefrontal and amygdala neuronal oscillations in male mice

Toya Okonogi ®[1], Nahoko Kuga[2], Musashi Yamakawa[2], Tasuku Kayama[2], Yuji Ikegaya ®[1,3,4] & Takuya Sasaki ®[1,2] ✉

The vagus nerve crucially affects emotions and psychiatric disorders. However, the detailed neurophysiological dynamics of the vagus nerve in response to emotions and its associated pathological changes remain unclear. In this study, we demonstrated that the spike rates of the cervical vagus nerve change depending on anxiety behavior in an elevated plus maze test, and these changes were eradicated in stress-susceptible male mice. Furthermore, instantaneous spike rates of the vagus nerve were negatively and positively correlated with the power of 2–4 Hz and 20–30 Hz oscillations, respectively, in the prefrontal cortex and amygdala. The oscillations also underwent dynamic changes depending on the behavioral state in the elevated plus maze, and these changes were no longer observed in stress-susceptible and vagotomized mice. Chronic vagus nerve stimulation restored behavior-relevant neuronal oscillations with the recovery of altered behavioral states in stress-susceptible mice. These results suggested that physiological vagal-brain communication underlies anxiety and mood disorders.

The vagus nerve (VN) plays a pivotal role in the communication between the peripheral organs and the brain. The afferent VN transmits ascending information regarding the internal physiological states of the visceral organs to the central nervous system, known as the so-called interoception[1,2]. Vagal interoceptive signals have a substantial effect on emotional states. For example, alterations in vagal signals (e.g., changes in gut microbiota or vagotomy) induce increased anxiety and depression-like behavior[3–8]. Furthermore, proper adjustment of vagal signals by vagus nerve stimulation (VNS) ameliorates treatment-resistant depression in humans[9–13] and induces anxiolytic[14,15] and anti-depressant effects[16,17] in rodents. These studies supported the idea that vagal interoceptive signals serve as a fundamental basis for sustaining emotional states.

While the importance of vagal-brain interactions in emotional functions has been widely recognized, it remains unclear how VN activity undergoes dynamic changes in relation to the anxiety states of individuals and how such a relationship between VN activity and anxiety is pathologically altered in mental disorders. Moreover, the neurophysiological mechanisms underlying VN-related anxiety and mental disorders remain largely elusive. The prefrontal cortex (PFC) and amygdala (AMY) have been suggested to be the principal brain regions associated with anxiety[18–20] and their interregional coordination of neuronal oscillations modulate anxiogenic behavior[21–25]. The importance of the VN in anxiety implies that these anxiety-related brain activity patterns may be crucially supported by the VN. In this study, we addressed these physiological issues by simultaneously recording the VN activity and brain local field potential (LFP) signals in naïve, stress-resilient, stress-susceptible, and vagotomized mice.

Finally, although VNS is an effective therapeutic strategy for mood disorders in both humans and animal models, its actual physiological

[1]Laboratory of Chemical Pharmacology, Graduate School of Pharmaceutical Sciences, The University of Tokyo, Tokyo 113–0033, Japan. [2]Department of Pharmacology, Graduate School of Pharmaceutical Sciences, Tohoku University, 6-3 Aramaki-Aoba, Aoba-Ku, Sendai 980-8578, Japan. [3]Institute for AI and Beyond, The University of Tokyo, Tokyo 113-0033, Japan. [4]Center for Information and Neural Networks, National Institute of Information and Communications Technology, Suita City, Osaka 565-0871, Japan. ✉e-mail: takuya.sasaki.b4@tohoku.ac.jp

mechanisms are largely unknown. In this study, we examined the influence of VNS on anxiety-related behavior and anxiety-relevant PFC-AMY LFP patterns in stress-susceptible mice. Overall, our results provided physiological insights into the dynamic cooperation between the VN and PFC-AMY circuits in anxiety and mental disorders.

## Results

### Reduced VN spikes during quiescent periods in stress-susceptible mice

C57BL/6J male mice were subjected to 10-min social defeat (SD) stress from CD-1 male mice that were screened as aggressor mice for 10 days (Fig. 1a, days 1–10). The stress susceptibility of mice after SD stress was assessed using the social interaction (SI) test (Fig. 1a, day 11). Of the 36 mice that received SD stress, 20 mice had SI ratios less than 1 and were classified as stress-susceptible mice, whereas 16 mice had SI ratios more than 1 and were classified as stress-resilient mice (Fig. 1b). To confirm whether these stress-induced phenotypes persist up to a maximum of three weeks later when electrophysiological recordings

and VNS were conducted, mice were exposed to 10-day SD stress, tested in a SI test on the next day (SI test 1), housed in their cages for more than three weeks, and again tested in a SI test (SI test 2) (Fig. 1c; $n = 15$ mice). We confirmed that the majority of the mice that were identified as stress-susceptible and stress-resilient phenotypes on the next day after SD stress continued to show the same phenotypes three weeks later, as verified by a significant positive correlation of the SI ratios between these two periods ($R = 0.66$, $P = 0.011$). This long-lasting effect allowed us to monitor stress-related electrophysiological activity for at least 3 weeks.

After identifying their stress susceptibility, the mice were implanted with a cuff-shaped electrode on the left VN (Fig. 1a, day 12; Fig. 1d, e), an EMG electrode on the dorsal neck muscle, and LFP electrodes on the PFC and AMY (Fig. 2b). After recovery from surgery for a week, electrophysiological recordings were performed when the mice were kept before starting a behavioral test in a rest box; this period was termed as "rest period" (Fig. 1a, day 20). VN signals were composed of complex compound extracellular signals from a

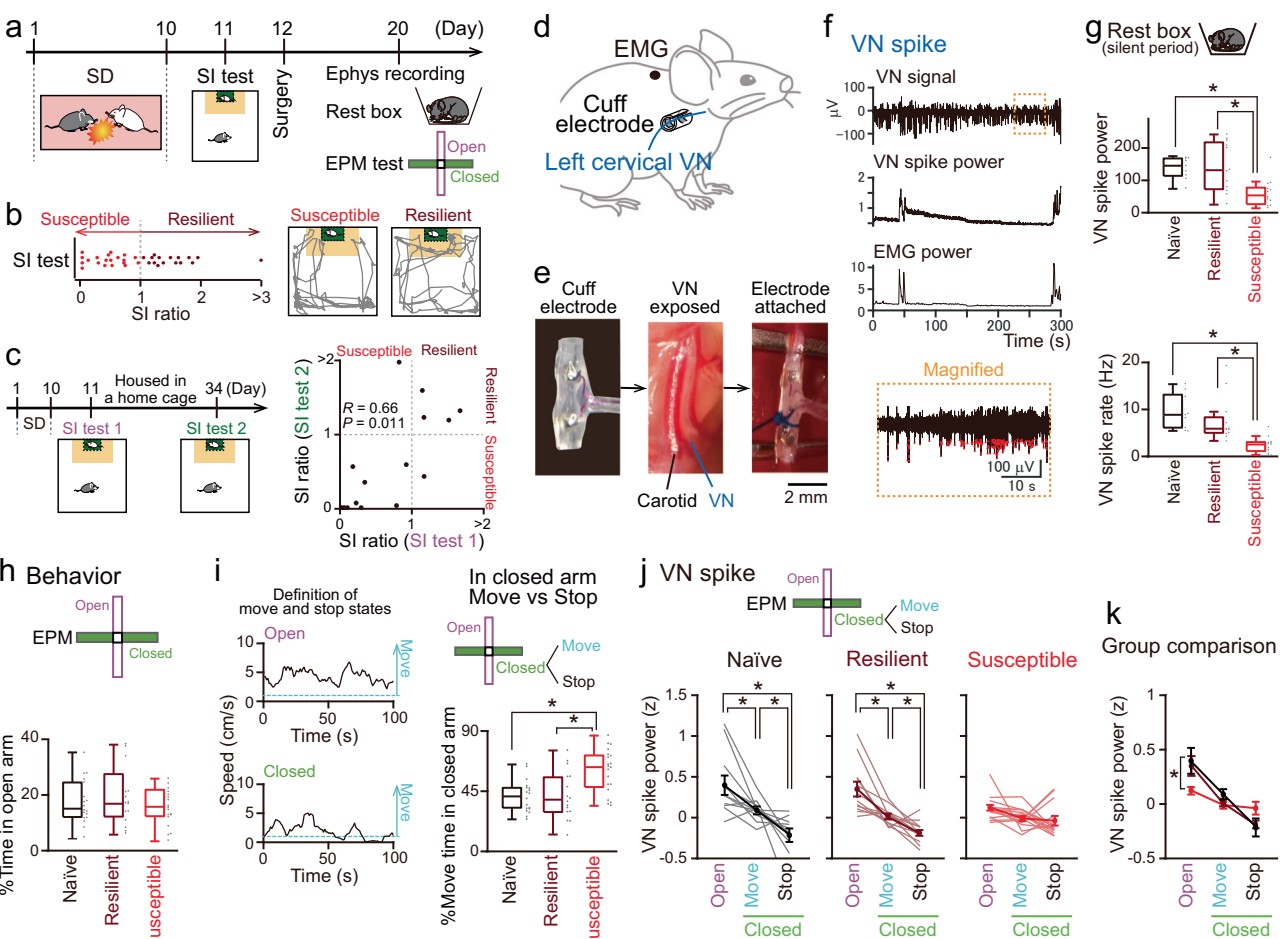

**Fig. 1 | SD stress attenuates VN spikes. a** Experimental timeline. **b** SI ratios ($n = 36$ mice) and representative trajectories in a SI test. **c** SI ratios at two time points ($n = 15$ mice). $R = 0.66$, $P = 0.011$, two-sided Pearson's Correlation. **d, e** Schematic illustration of VN recordings with a cuff-shaped electrode. Similar electrodes used for 45 mice. **f** A filtered VN trace, its power, and EMG power during a rest period. The orange region is magnified (bottom; dots represent spikes). **g** VN spike power and rates during quiescent periods ($n = 9$, 12, and 12 mice). Box plots show center line as median, box limits as the 25th percentile and the 75th percentile, whiskers as minimum to maximum values that are not outliers. power: naïve vs susceptible, *$P = 0.012$, resilient vs susceptible, *$P = 0.0044$; rate: naïve vs susceptible, *$P = 3.9 \times 10^{-4}$, resilient vs susceptible, *$P = 0.0040$, two-sided Tukey's test. **h** The percentages of open arms ($n = 18$, 16, and 20 mice). Box plots show center line as

median, box limits as the 25th percentile and the 75th percentile, whiskers as minimum to maximum values that are not outliers. **i** (Left) Moving speed. (Right) Same as **h** but for the percentage of move states in closed arms. naïve vs susceptible, *$P = 0.0012$, resilient vs susceptible, *$P = 0.0029$, two-sided Tukey's test. **j** VN spike power in the arms ($n = 11$, 12, and 14 mice). Data are presented as mean ± SEM. naïve: open vs closed move, $t_{10} = 3.10$, *$P = 0.034$; open vs closed stop, $t_{10} = 4.16$, *$P = 0.0058$; closed move vs closed stop, $t_{10} = 2.98$, *$P = 0.041$; resilient: open vs closed move, $t_{11} = 3.85$, *$P = 0.0081$; open vs closed stop, $t_{11} = 5.74$, *$P = 3.9 \times 10^{-4}$; closed move vs closed stop, $t_{11} = 3.87$, *$P = 0.0078$, two-sided paired $t$-test followed by Bonferroni correction. **k** Across-group comparisons. *$P = 0.035$, two-sided Tukey's test. Source data are provided as a Source Data file.

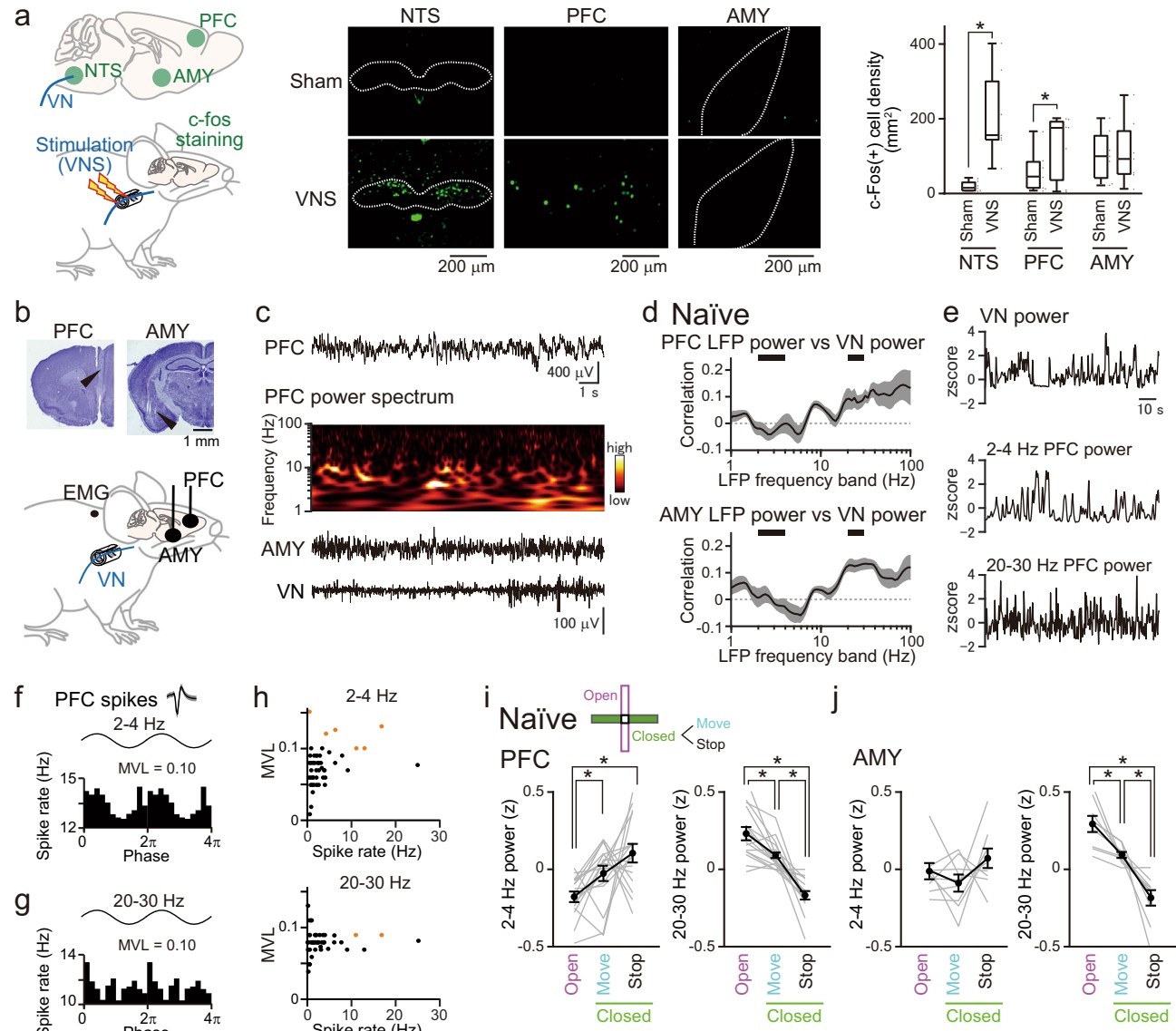

**Fig. 2 | VN activity and PFC-AMY oscillations. a** Images of c-Fos expressions in the NTS, PFC, and AMY, after VNS (left) and the percentages of c-Fos-positive neurons (right; $n = 7$ and 7 mice). Box plots show center line as median, box limits as the 25th percentile and the 75th percentile, whiskers as minimum to maximum values that are not outliers. NTS: $t_{12} = 14.86$, $*P = 4.3 \times 10^{-9}$; PFC: $t_{12} = 3.53$, $*P = 0.0042$; AMY: $t_{12} = 0.98$, $P = 0.35$, two-sided Student's $t$-test from bootstrapped datasets. **b** Histological confirmation of electrodes. Similar recordings were obtained from the PFC and AMY in 63 and 51 mice, respectively. **c** A PFC LFP trace, its wavelet spectrum, an AMY LFP trace, and a VN trace. **d** Correlations between VN power and PFC or AMY power against each frequency band in naïve mice ($n = 9$ and 9 mice). Shaded areas represent SEM. Black bars represent 2–4 Hz and 20–30 Hz bands. **e** Changes in VN power and PFC power. **f, g** PFC neuronal firing with 2–4 Hz and

20–30 Hz oscillations. **h** MVLs of PFC neurons for 2–4 Hz and 20–30 Hz oscillations against their firing rates ($n = 47$ neurons from 6 mice). Orange dots show significance. **i** PFC 2–4 Hz and 20–30 Hz power on the EPM ($n = 16$ mice). Data are presented as mean ± SEM. 2–4 Hz: open vs closed move, $t_{15} = 2.87$, $*P = 0.035$; open vs closed stop, $t_{15} = 3.56$, $*P = 0.0086$; 20–30 Hz: open vs closed move, $t_{15} = 3.89$, $*P = 0.0044$; open vs closed stop, $t_{15} = 6.24$, $*P = 4.7 \times 10^{-5}$; closed move vs closed stop, $t_{15} = 6.07$, $*P = 6.4 \times 10^{-5}$; two-sided paired $t$-test followed by Bonferroni correction. **j** Same as **i** but for AMY ($n = 9$ mice). open vs closed move, $t_8 = 3.73$, $*P = 0.017$; open vs closed stop, $t_8 = 6.32$, $*P = 6.8 \times 10^{-4}$; closed move vs closed stop, $t_8 = 4.13$, $*P = 0.0099$; two-sided paired $t$-test followed by Bonferroni correction. Source data are provided as a Source Data file.

considerable number of VN fibers (Fig. 1f, top). Based on anatomical observations that 75–90% of the fibers in the cervical VN are afferent fibers[26–28], the majority of VN signals are considered to represent afferent fiber activity. When mice moved with large EMG amplitudes, both the amplitude and frequency of the compound VN spikes became considerably high[29], making it impossible to isolate them into individual spike units. From these movement periods (e.g., on an elevated plus maze [EPM] test), we only computed the root-mean-square of the VN signals (Fig. 1f, middle) as VN spike power, which were considered to represent the overall information transmitted through the VN from the peripheral organs to the brain. On the other hand, during quiescent

periods with smaller EMG amplitudes during rest periods, VN signals were still composed of compound spikes, but individual extracellular spike-like signals appeared to be more prominent (magnified in Fig. 1f bottom). From these periods, in addition to VN spike power, we detected the timing of individual spike units when the amplitude of the negative deflection of the filtered VN signals exceeded a threshold (approximately $5 \times SD$ below the average) and estimated overall VN spike rates. We first computed both VN spike power and VN spike rates when mice were quiescent in a home cage before an EPM test (Fig. 1g). Naïve and resilient mice showed larger VN spike power and higher VN spike rates than stress-susceptible mice (Fig. 1g; $n = 9$ naïve, 12 resilient,

and 12 susceptible mice; spike power: naïve vs susceptible, $P = 0.012$; resilient vs susceptible, $P = 0.0044$, two-sided Tukey's test. $F_{2,32} = 7.40$, $P = 0.0024$, one-way ANOVA; spike rate: naïve vs susceptible, $P = 3.9 \times 10^{-4}$, resilient vs susceptible, $P = 0.0040$, two-sided Tukey's test. $F_{2,32} = 11.01$, $P = 3.0 \times 10^{-4}$, one-way ANOVA). In contrast, no significant differences in VN spike power and VN spike rates were found between the naïve and stress-resilient mice (spike power: $P = 0.99$; spike rate: $P = 0.50$, two-sided Tukey's test. $F_{2,32} = 11.01$, $P = 3.0 \times 10^{-4}$, one-way ANOVA). These results demonstrate that intrinsic VN activity during quiescent periods is reduced in stress-susceptible mice. In all mouse groups, these parameters of VN activity during a rest period before the EPM test (termed the pre-rest period) were not significantly different from those observed during a rest period after the EPM test (termed the post-rest period) (Supplementary Fig. 1). These results suggest that basal VN activity during quiescent periods is not prominently affected by experiences of anxiety-related environments in the EPM.

### Alterations in behavioral patterns and behavior-relevant VN spike power in the EPM test of stress-susceptible mice

Chronic SD stress in rodents alters anxiety behaviors[30–33]. After recording the VN signals during rest periods, the mice were subjected to an EPM test (Fig. 1h). No significant differences in the percentage of open arms were found among the three mouse groups ($n = 18$ naïve, 16 resilient, and 20 susceptible mice; $F_{2,53} = 0.60$, $P = 0.55$, one-way ANOVA; naïve vs resilient, $P = 0.58$, naïve vs susceptible, $P = 0.99$, resilient vs susceptible, $P = 0.63$, two-sided Tukey's test), indicating that the effect of SD stress was not evident in the conventional measure of an EPM test (e.g., time in open/closed arms), as reported in several studies[34,35]. A previous study demonstrated that physiological brain and peripheral organ states are not consistent but vary considerably in closed arms in an EPM test[36]. Based on this observation, we classified behavioral patterns into move and stop states with running speeds of more than and less than 1 cm/s, respectively (Fig. 1i, left). Irrespective of the mouse groups, virtually all (96.0%) periods in the open arms were classified as move states (Fig. 1i, left top). Thus, we equivalently examined all periods in the open arms. In contrast, 41.2% of the periods in the closed arms were classified as move states in naïve mice. We thus separately analyzed move and stop states in the closed arms. Compared to naïve mice, stress-susceptible mice (but not resilient mice) showed significantly longer move states in the closed arms (Fig. 1i, right; $F_{2,53} = 9.07$, $P = 0.00040$, one-way ANOVA; naïve vs resilient, $P = 0.99$, naïve vs susceptible, $P = 0.0012$, resilient vs susceptible, $P = 0.0029$, two-sided Tukey's test). These results indicate that the effect of our SD stress paradigm in stress-susceptible mice was evident from differences in move/stop states in the closed arms.

After identifying the behavioral changes in the EPM test, we compared the overall VN spike power between the open/closed arms and move/stop states in each mouse group. In naïve mice, VN spike power in the open arms was significantly higher than in the closed arms and VN spike power during move states in the closed arms was significantly higher than during stop states in the closed arms (Fig. 1j, left; $n = 11$ mice; open vs closed move, $t_{10} = 3.10$, $P = 0.034$; open vs closed stop, $t_{10} = 4.16$, $P = 0.0058$; closed move vs closed stop, $t_{10} = 2.98$, $P = 0.041$, two-sided paired $t$-test followed by Bonferroni correction). Similar significant differences in VN spike power were observed in stress-resilient mice (Fig. 1j, middle; $n = 12$ mice; open vs closed move, $t_{11} = 3.85$, $P = 0.0081$; open vs closed stop, $t_{11} = 5.74$, $P = 3.9 \times 10^{-4}$; closed move vs closed stop, $t_{11} = 3.87$, $P = 0.0078$, two-sided paired $t$-test followed by Bonferroni correction). These results demonstrate that VN spike intensity undergoes dynamic changes depending on the arm and behavioral state in naïve and stress-resilient mice. Conversely, no significant differences in VN spike power were found across these arms and behavioral states in stress-susceptible mice (Fig. 1j, right; $n = 14$ mice; open vs closed move, $t_{13} = 2.13$, $P = 0.16$; open vs closed

stop, $t_{13} = 1.99$, $P = 0.21$; closed move vs closed stop, $t_{13} = 0.63$, $P > 0.99$; two-sided paired $t$-test followed by Bonferroni correction). Across-group comparisons showed that VN spike power in the open arms in naïve mice was significantly higher than that in stress-susceptible mice but not in stress-resilient mice (Fig. 1k; open: $F_{2,36} = 3.95$, $P = 0.029$, one-way ANOVA; naïve vs resilient, $P = 0.86$, naïve vs susceptible, $P = 0.035$, resilient vs susceptible, $P = 0.10$, two-sided Tukey's test). Taken together, these results suggest that VN spike patterns were not properly adjusted in response to changes in anxiety-related behavioral states in stress-susceptible mice.

Some of the mice showed intermediate SI ratios close to 1 (i.e. the threshold to define stress susceptibility) and they may influence the statistical differences in VN spike patterns. We further defined strongly resilient and strongly susceptible mouse groups by evenly dividing the stress-resilient and stress-susceptible mice into two subgroups. A subgroup with higher SI ratios (>1.48) in the resilient mice was classified as strongly resilient mice ($n = 8$ mice), whereas a subgroup with lower SI ratios (<0.44) in the susceptible mice was classified as strongly susceptible mice ($n = 10$ mice). Similar to the results in Fig. 1g, VN spike rates during quiescent periods were significantly higher in naïve mice, compared with strongly susceptible mice, but not with strongly resilient mice (Supplementary Fig. 2b; $n = 9$, 6, and 5 mice; $F_{2,19} = 12.38$, $P = 0.00050$, one-way ANOVA; naïve vs strongly resilient, $P = 0.077$, naïve vs strongly susceptible, $P = 0.00030$, strongly resilient vs strongly susceptible, $P = 0.0545$, two-sided Tukey's test). Similar to the behavioral patterns in Fig. 1h, i, no significant differences in the percentages of open arms were found among the three mouse groups (Supplementary Fig. 2c, left; $n = 18$ naïve, 8 strongly resilient, and 10 strongly susceptible mice; $F_{2,35} = 0.10$, $P = 0.90$, one-way ANOVA; naïve vs strongly resilient, $P = 0.99$, naïve vs strongly susceptible, $P = 0.93$, strongly resilient vs strongly susceptible, $P = 0.91$, two-sided Tukey's test) and strongly susceptible mice specifically showed significantly longer move states in the closed arms, compared to naïve mice (Supplementary Fig. 2c, right; $F_{2,35} = 3.78$, $P = 0.033$, one-way ANOVA; naïve vs strongly resilient, $P = 0.91$, naïve vs strongly susceptible, $P = 0.028$, strongly resilient vs strongly susceptible, $P = 0.16$, two-sided Tukey's test). Moreover, similar to the VN spike patterns in Fig. 1j, strongly resilient mice showed significant differences in VN spike power between the open arms or move states in the closed arms and stop states in the closed arms (Supplementary Fig. 3a, leftmost; $n = 6$ mice; open vs closed move, $t_5 = 3.33$, $P = 0.060$; open vs closed stop, $t_5 = 5.83$, $P = 0.0063$; closed move vs closed stop, $t_5 = 6.00$, $P = 0.0055$, two-sided paired $t$-test followed by Bonferroni correction), whereas strongly susceptible mice did not show such significant differences (Supplementary Fig. 3b, leftmost; $n = 5$ mice; open vs closed move, $t_4 = 1.22$, $P = 0.87$; open vs closed stop, $t_4 = 3.25$, $P = 0.095$; closed move vs closed stop, $t_4 = 1.28$, $P = 0.81$, two-sided paired $t$-test followed by Bonferroni correction). These results confirm that the classifications into strongly resilient and strongly susceptible mouse groups led to the same results in behavioral and VN spike analyses as observed from the classification between resilient and susceptible mouse groups.

### VN spike patterns are correlated with PFC-AMY oscillations

The PFC and AMY are crucial brain regions in anxiety[22,24,37]. Next, we tested whether these anxiety-related brain regions were affected by VN activity. The mice were subjected to electrical VNS (amplitude = 0.8 mA, width = 0.1 ms, frequency = 20 Hz, number of pulses = 600, total duration = 30 s) every 1 min for 3 h through cuff-shaped electrodes attached to the left cervical VN, and the expression levels of c-Fos, an immediate early gene as a marker of neuronal activation, were measured in these brain regions 1.5 h after the VNS (Fig. 2a). Compared to sham-operated mice without VNS, the mice with VNS exhibited significantly larger proportions of c-Fos-positive neurons in the nucleus tractus solitarius (NTS) (Fig. 2b; $n = 7$ VNS-stimulated and 7 sham-operated mice, $t_{12} = 14.86$, $P = 4.3 \times 10^{-9}$, Student's $t$-test from

bootstrapped datasets), a primary brain region that receives direct inputs from the afferent VN, confirming the successful activation of the VN. Under these conditions, a significant increase in the proportion of c-Fos-positive neurons was detected in the PFC ($t_{12} = 3.53$, $P = 0.0042$) but not in the AMY ($t_{12} = 0.98$, $P = 0.35$). Based on this histological confirmation, we mainly analyzed PFC LFP signals, whereas the same analysis was applied to AMY LFP signals (Fig. 2b, c). The LFP signals were converted into power spectra by wavelet analysis (Fig. 2c). Temporal activity relationships between the VN and these brain regions were quantified as correlation coefficients between instantaneous VN spike power and LFP power in the frequency bands from 1 to 100 Hz (Fig. 2d). In 9 naïve mice tested, 6 mice showed significantly negative correlations between PFC 2–4 Hz LFP power and VN spike power and 7 mice showed significantly positive correlations between PFC 20–30 Hz LFP power and VN spike power (Fig. 2d, top; representative LFP traces shown in Fig. 2e). The same significantly negative correlations were observed from AMY 2–4 Hz LFP signals in 4 mice out of 9 naïve mice tested, whereas significantly positive correlations were observed from AMY 20–30 Hz LFP signals in all the 9 mice tested (Fig. 2d, bottom). These results suggest that LFP oscillations in these frequency bands are candidate signals associated with VN spike activity in the PFC-AMY circuits.

Next, we examined whether oscillatory cycles in PFC LFP oscillations are phase-locked to spikes of individual PFC neurons during rest periods. All recorded PFC cells were included in the analyses, irrespective of the putative cell type. An example PFC neuron shown in Fig. 2f, g exhibited spike rate changes corresponding to the altering phases in the 2–4 Hz and 20–30 Hz oscillations, respectively. The degree of spike phase locking was quantified using the mean vector length (MVL). The significance of an MVL was defined based on shuffled datasets ($P < 0.001$). Of the 47 PFC neurons from the six tested mice, six (12.8%) and two (4.3%) neurons showed significant MVLs for the 2–4 Hz and 20–30 Hz oscillations, respectively (orange dots in Fig. 2h). These results suggest that a subset of PFC neurons is prominently entrained by PFC LFP oscillations in these frequency bands.

The findings of the behavior-dependent VN spike patterns in the EPM test and the correlational power changes between VN spike power and 2–4 Hz or 20–30 Hz LFP power in the PFC-AMY circuits suggest that the LFP power in these frequency bands undergoes changes with behavioral states in the EPM test. To confirm this possibility, the instantaneous LFP power on the EPM was z-scored in each frequency band for each mouse. Overall, z-scored 2–4 Hz and 20–30 Hz PFC power was significantly lower and higher in the open arms, respectively, than in the closed arms (Fig. 2i; $n = 16$ mice; 2–4 Hz: open vs closed move, $t_{15} = 2.87$, $P = 0.035$; open vs closed stop, $t_{15} = 3.56$, $P = 0.0086$; 20–30 Hz: open vs closed move, $t_{15} = 3.89$, $P = 0.0044$; open vs closed stop, $t_{15} = 6.24$, $P = 4.7 \times 10^{-5}$, two-sided paired $t$-test followed by Bonferroni correction). Furthermore, the z-scored 20–30 Hz power, but not 2–4 Hz power, in the PFC during move states was significantly higher than that during stop states in the closed arms (2–4 Hz: closed move vs closed stop, $t_{15} = 1.36$, $P = 0.58$; 20–30 Hz: closed move vs closed stop, $t_{15} = 6.07$, $P = 6.4 \times 10^{-5}$). Similar significant changes were also observed at 20–30 Hz but not 2–4 Hz, in the AMY (Fig. 2j; $n = 9$ mice; 2–4 Hz: open vs closed move, $t_8 = 1.03$, $P > 0.99$; open vs closed stop, $t_8 = 0.87$, $P > 0.99$; closed move vs closed stop, $t_8 = 1.46$, $P = 0.55$; 20–30 Hz: open vs closed move, $t_8 = 3.73$, $P = 0.017$; open vs closed stop, $t_8 = 6.32$, $P = 6.8 \times 10^{-4}$; closed move vs closed stop, $t_8 = 4.13$, $P = 0.0099$; two-sided paired $t$-test followed by Bonferroni correction). These results suggest that the 2–4 Hz and 20–30 Hz LFP patterns in the PFC circuit change dynamically with stress-sensitive anxiety behavior.

## PFC LFP patterns are disrupted in stress-susceptible mice

Next, we examined how anxiety-related PFC-AMY oscillations were altered in stress-resilient and stress-susceptible mice. Similar to the naïve mice, stress-resilient mice showed significantly higher 20–30 Hz LFP power in the PFC and AMY in the open arms and during move periods in the closed arms than during stop periods in the closed arms (Fig. 3a, b; $n = 10$ and 11 mice; PFC: open vs closed move, $t_9 = 2.14$, $P = 0.18$; open vs closed stop, $t_9 = 5.29$, $P = 0.0015$; closed move vs closed stop, $t_9 = 6.00$, $P = 6.0 \times 10^{-4}$; AMY: open vs closed move, $t_{10} = 0.87$, $P > 0.99$; open vs closed stop, $t_{10} = 4.73$, $P = 0.0024$; closed move vs closed stop, $t_{10} = 4.95$, $P = 0.0017$, two-sided paired $t$-test followed by Bonferroni correction). However, they did not show significant differences in 2–4 Hz power in these brain regions among the three behavioral states (PFC: open vs closed move, $t_9 = 1.24$, $P = 0.74$; open vs closed stop, $t_9 = 1.41$, $P = 0.57$; closed move vs closed stop, $t_9 = 0.10$, $P > 0.99$; AMY: open vs closed move, $t_{10} = 0.49$, $P > 0.99$; open vs closed stop, $t_{10} = 1.43$, $P = 0.55$; closed move vs closed stop, $t_{10} = 1.49$, $P = 0.50$, two-sided paired $t$-test followed by Bonferroni correction). Similar significant differences in PFC LFP patterns were observed when the analyses were restricted to strongly resilient mice (Supplementary Fig. 3a, right; $n = 7$ and 5 mice; 2–4 Hz: open vs closed move, $t_6 = 1.57$, $P = 0.50$; open vs closed stop, $t_6 = 1.12$, $P = 0.92$; closed move vs closed stop, $t_6 = 0.31$, $P > 0.99$; 20–30 Hz: open vs closed move, $t_6 = 2.95$, $P = 0.078$; open vs closed stop, $t_6 = 4.40$, $P = 0.014$; closed move vs closed stop, $t_6 = 4.45$, $P = 0.013$, two-sided paired $t$-test followed by Bonferroni correction.)

In contrast, stress-susceptible mice exhibited no significant differences in either 2–4 Hz or 20–30 Hz LFP power in the PFC and AMY among the open arms, move states in the closed arms, and stop states in the closed arms (Fig. 3c, d; $n = 15$ and 10 mice; PFC: 2–4 Hz: open vs closed move, $t_{14} = 2.56$, $P = 0.069$; open vs closed stop, $t_{14} = 1.28$, $P = 0.66$; closed move vs closed stop, $t_{14} = 0.69$, $P > 0.99$; 20–30 Hz: open vs closed move, $t_{14} = 0.29$, $P > 0.99$; open vs closed stop, $t_{14} = 1.78$, $P = 0.29$; closed move vs closed stop, $t_{14} = 1.88$, $P = 0.24$; AMY: 2–4 Hz: open vs closed move, $t_9 = 0.54$, $P > 0.99$; open vs closed stop, $t_9 = 0.78$, $P > 0.99$; closed move vs closed stop, $t_9 = 0.73$, $P > 0.99$; 20–30 Hz: open vs closed move, $t_9 = 1.15$, $P = 0.83$; open vs closed stop, $t_9 = 2.21$, $P = 0.16$; closed move vs closed stop, $t_9 = 2.56$, $P = 0.093$, two-sided paired $t$-test followed by Bonferroni correction). The same non-significant results for PFC LFP patterns were observed when the analyses were restricted to strongly susceptible mice (Supplementary Fig. 3b, right; $n = 10$ mice; 2–4 Hz: open vs closed move, $t_9 = 2.10$, $P = 0.20$; open vs closed stop, $t_9 = 1.51$, $P = 0.49$; closed move vs closed stop, $t_9 = 0.30$, $P > 0.99$; 20–30 Hz: open vs closed move, $t_9 = 0.40$, $P > 0.99$; open vs closed stop, $t_9 = 0.13$, $P > 0.99$; closed move vs closed stop, $t_9 = 0.38$, $P > 0.99$, two-sided paired $t$-test followed by Bonferroni correction). Across-group comparisons showed that naïve and stress-resilient mice had significantly higher and lower PFC power at 20–30 Hz in the open arms and during stop states in the closed arms, respectively, than stress-susceptible mice (Fig. 3e, f; PFC: 20–30 Hz: open: $F_{2,40} = 5.40$, $P = 0.0086$, one-way ANOVA; naïve vs resilient, $P = 0.64$, naïve vs susceptible, $P = 0.048$, resilient vs susceptible, $P = 0.011$; closed stop: 20–30 Hz: $F_{2,40} = 5.17$, $P = 0.010$, one-way ANOVA; naïve vs resilient, $P = 1.00$, naïve vs susceptible, $P = 0.016$, resilient vs susceptible, $P = 0.042$, two-sided Tukey's test). Taken together, these results demonstrate that anxiety-related LFP power changes in the PFC-AMY circuits were specifically disrupted in stress-susceptible mice, but not in stress-resilient mice, which is consistent with the results of VN power changes.

## Vagotomy induces anxiety-related behavior and disrupts LFP power changes in PFC-AMY circuits

The relationship between VN activity and PFC-AMY oscillations and their disruption in stress-susceptible mice with decreased VN spike activity implies that the VN plays a crucial role in the creation of anxiety-related LFP patterns in PFC-AMY circuits. To test this hypothesis, the left cervical VN was physically cut (vagotomy), and PFC and AMY LFP signals were recorded from vagotomized mice in the EPM

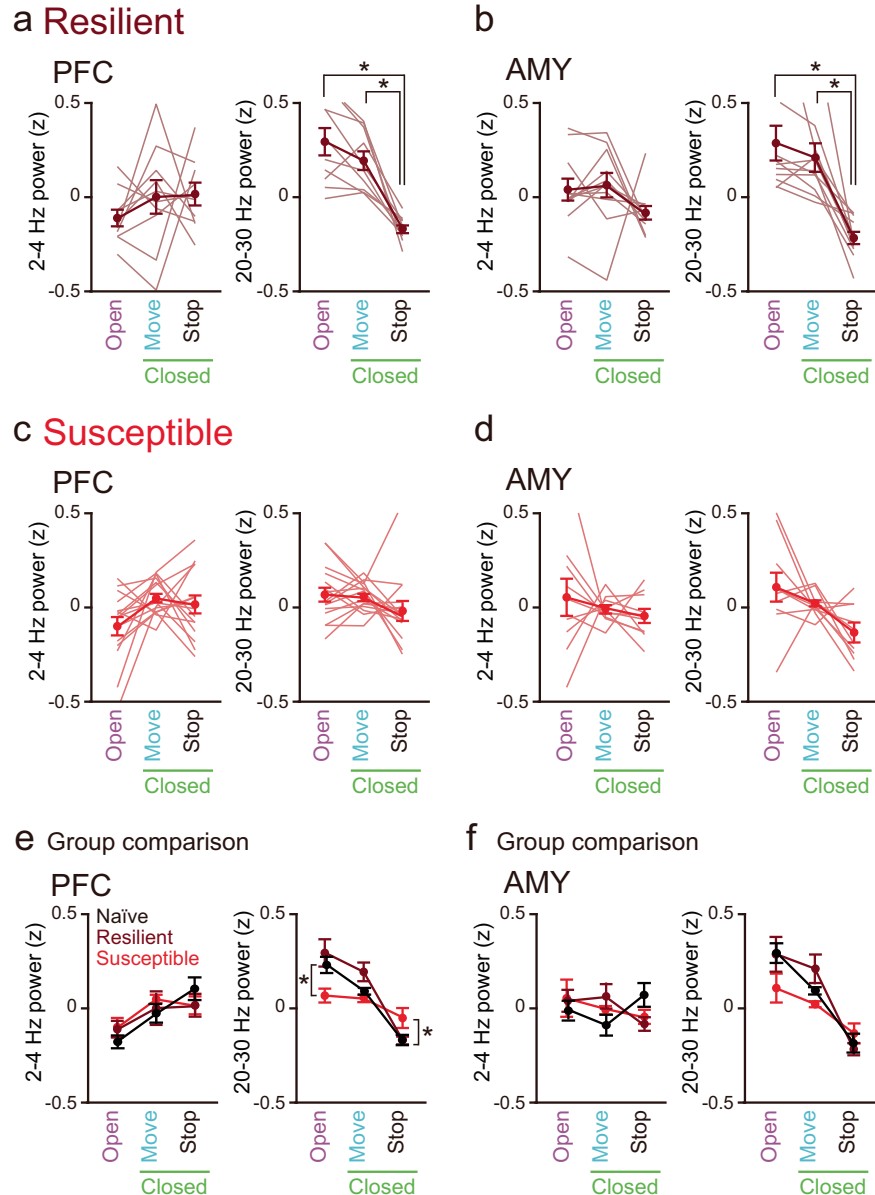

**Fig. 3 | Disruption of anxiety-related PFC-AMY LFP patterns in stress-susceptible mice. a** Comparisons of PFC 2–4 Hz and 20–30 Hz power among the open arms, move states in the closed arms, and stop states in the closed arms in stress-resilient mice ($n = 10$ mice). Each thin line represents each mouse. Data are presented as mean ± SEM. 20–30 Hz: open vs closed stop, $t_9 = 5.29$, *$P = 0.0015$; closed move vs closed stop, $t_9 = 6.00$, *$P = 6.0 \times 10^{-4}$, two-sided paired $t$-test followed by Bonferroni correction. **b** Same as a but for AMY power ($n = 11$ mice). 20–30 Hz: open vs closed stop, $t_{10} = 4.73$, *$P = 0.0024$; closed move vs closed stop, $t_{10} = 4.95$, *$P = 0.0017$; two-sided paired $t$-test followed by Bonferroni correction. **c, d** Same as **a**, **b** but for stress-susceptible mice ($n = 15$ and 10 mice). **e** Across-group comparisons of PFC power shown in Figs. 2i and 3a, c. 20–30 Hz: open: naïve vs susceptible, *$P = 0.048$, resilient vs susceptible, *$P = 0.011$; closed stop: naïve vs susceptible, *$P = 0.016$, resilient vs susceptible, *$P = 0.042$, two-sided Tukey's test. **f** Across-group comparisons of the datasets of AMY power shown in Figs. 2j and 3b, d. Source data are provided as a Source Data file.

test (Fig. 4a). Although no significant differences in the percentage of open arms were found between naïve and vagotomized mice (Fig. 4b, left; $n = 18$ naïve and 11 vagotomized mice; $t_{27} = 1.68$, $P = 0.10$, two-sided Student's $t$-test), the vagotomized mice showed significantly longer move states than the naïve mice (Fig. 4b, right; $t_{27} = 3.20$, $P = 0.0035$, two-sided Student's $t$-test). Notably, these behavioral phenotypes were similar to those observed in stress-susceptible mice. Similar to the stress-susceptible mice, the vagotomized mice exhibited no significant changes in 2–4 Hz and 20–30 Hz power in the PFC and AMY among the open arms, move states in the closed arms, and stop states in the closed arms (Fig. 4c, d; $n = 11$ and 11 mice; PFC: 2–4 Hz: open vs closed move, $t_{10} = 1.22$, $P = 0.75$; open vs closed stop, $t_{10} = 2.64$, $P = 0.061$; closed move vs closed stop, $t_{10} = 2.45$, $P = 0.10$; 20–30 Hz:

open vs closed move, $t_{10} = 1.31$, $P = 0.66$; open vs closed stop, $t_{10} = 2.19$, $P = 0.16$; closed move vs closed stop, $t_{10} = 2.50$, $P = 0.094$; AMY: 2–4 Hz: open vs closed move, $t_{10} = 0.75$, $P > 0.99$; open vs closed stop, $t_{10} = 2.60$, $P = 0.080$; closed move vs closed stop, $t_{10} = 2.29$, $P = 0.14$; 20–30 Hz: open vs closed move, $t_{10} = 0.97$, $P > 0.99$; open vs closed stop, $t_{10} = 1.68$, $P = 0.37$; closed move vs closed stop, $t_{10} = 1.29$, $P = 0.68$; two-sided paired $t$-test followed by Bonferroni correction). These results suggest that VN activity is necessary to create dynamic changes in PFC-AMY 2–4 Hz and 20–30 Hz powers. Taken all, the altered behavioral patterns and unchanged PFC and AMY LFP patterns in vagotomized mice were similar to those observed in stress-susceptible mice, implying that stress-susceptible phenotypes are mediated by the same mechanism, that is, the disruption of VN activity.

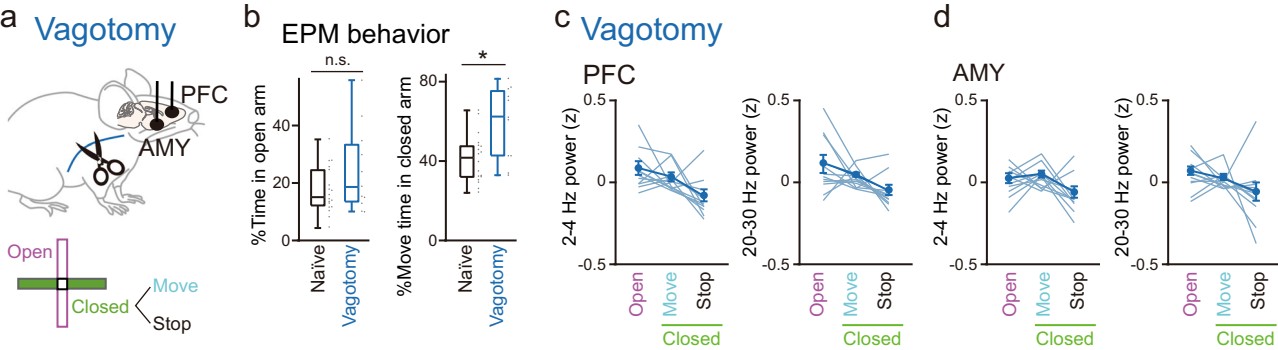

**Fig. 4 | Vagotomy disrupts anxiety-related PFC-AMY LFP patterns. a** LFP signals were recorded from mice with vagotomy. **b** The percentage of time spent in the open arms (left; $t_{27} = 1.68$, $P = 0.10$, two-sided Student's $t$-test) and the percentage of move states in the closed arms (right; $t_{27} = 3.20$, $*P = 0.0035$, two-sided Student's $t$-test) to the total recording time in vagotomized mice ($n = 11$ mice). Box plots show center line as median, box limits as the 25th percentile and the 75th percentile, whiskers as minimum to maximum values that are not outliers. **c** Comparisons of PFC 2–4 Hz and 20–30 Hz power among the open arms, move states in the closed arms, and stop states in the closed arms in vagotomized mice ($n = 11$ mice). Each thin line represents each mouse. Data are presented as mean ± SEM. **d** Same as c but for AMY power ($n = 11$ mice). Source data are provided as a Source Data file.

## VNS restores anxiety-related behavior and PFC LFP power changes

All observations from the various mouse types suggest that increased VN excitability in stress-susceptible mice restores anxiety-related behavioral states and dysfunctional PFC LFP patterns. To address this idea, the mice showing stress-susceptible phenotypes were specifically selected after 10 days of SD stress (Fig. 5a; $n = 18$ mice) and were implanted with an electrode on the left VN (used for both stimulation and recording), an EMG electrode, and LFP electrodes into the PFC and AMY. After recovery from the surgery, the mice were subjected to VNS daily with the same stimulation parameters as in Fig. 2a for 1–2 weeks and were grouped into the "susceptible + VNS mouse" group. The next day, electrophysiological recordings and behavioral tests were performed. In the susceptible + VNS group, SI ratios after VNS were significantly higher than those observed before VNS (Fig. 5b; $n = 18$ and 10 mice; $P = 2.3 \times 10^{-4}$, Mann–Whitney $U$ test), suggesting that VNS was effective in restoring stress-induced social interaction deficits. Since we demonstrated a significant positive correlation of SI ratios between one day and three weeks after the 10-days SD stress in mice without VNS (Fig. 1c), the observed effect of VNS is not simply explained by the spontaneous recovery of behavioral phenotypes for 3 weeks after SD stress. In the EPM test, the susceptible + VNS mouse group showed significantly shorter move states in the closed arms than in the susceptible group without VNS (Fig. 5c, right; $n = 20$ and 18 mice, $t_{37} = 3.04$, $P = 0.0044$, two-sided Student's $t$-test), whereas the percentages of the open arms did not significantly differ between the two groups (Fig. 5c, left; $t_{37} = 0.043$, $P = 0.97$, two-sided Student's $t$-test). The restored behavioral phenotypes in the susceptible + VNS mouse group were similar to those observed in naïve mice (Fig. 1h, i). Consistently, the VN spike power in the open arms and during move states in the closed arms was significantly higher than that during stop states in the closed arms in the susceptible + VNS mouse group (Fig. 5d; $n = 7$ mice; open vs closed move, $t_6 = 3.07$, $P = 0.066$; open vs closed stop, $t_6 = 5.88$, $P = 0.0032$; closed move vs closed stop, $t_6 = 7.72$, $P = 7.4 \times 10^{-4}$; two-sided paired $t$-test followed by Bonferroni correction). Overall, similar to the naïve mice as observed in Fig. 1j, the susceptible + VNS mouse group showed significantly higher VN spike power in the open arms than the stress-susceptible mice (Fig. 5e; open: $t_{19} = 4.19$, $P = 5.0 \times 10^{-4}$: closed stop: $t_{19} = 2.61$, $P = 0.017$, two-sided Student's $t$-test). Taken together, these results demonstrate that VNS applied to stress-susceptible mice restored both basal VN spike power and behavior-relevant VN spike patterns, in accordance with the restoration of their behavioral patterns in the EPM test.

For the LFP patterns, z-scored 20–30 Hz power in the PFC and AMY in the open arms was significantly higher than that in the closed arms in the susceptible+VNS mouse group (Fig. 5f, g; $n = 11$ and 10 mice; PFC: open vs closed move, $t_{10} = 5.12$, $P = 0.0013$; open vs closed stop, $t_{10} = 6.09$, $P = 3.5 \times 10^{-4}$; closed move vs closed stop, $t_{10} = 4.76$, $P = 0.0023$; AMY: open vs closed move, $t_9 = 2.16$, $P = 0.18$; open vs closed stop, $t_9 = 7.87$, $P = 7.6 \times 10^{-5}$; closed move vs closed stop, $t_9 = 5.26$, $P = 0.0016$, two-sided paired $t$-test followed by Bonferroni correction), as observed in the naïve mice. On the other hand, no significant differences in 2–4 Hz PFC and AMY power were observed across the behavioral states in the susceptible + VNS mouse group (Fig. 5f, g; $n = 11$ and 10 mice; PFC: open vs closed move, $t_{10} = 2.39$, $P = 0.11$; open vs closed stop, $t_{10} = 1.61$, $P = 0.42$; closed move vs closed stop, $t_{10} = 0.28$, $P > 0.99$; AMY: open vs closed move, $t_9 = 0.10$, $P > 0.99$; open vs closed stop, $t_9 = 0.91$, $P > 0.99$; closed move vs closed stop, $t_9 = 1.18$, $P = 0.80$, two-sided paired $t$-test followed by Bonferroni correction). Overall, similar to the naïve mice observed in Fig. 3e, the susceptible + VNS mouse group showed significantly higher and lower 20–30 Hz PFC power in the open arms and during stop states in the closed arms, respectively, than the stress-susceptible mice (Fig. 5h, i; PFC: open: $t_{24} = 3.47$, $P = 0.020$: closed stop: $t_{24} = 2.18$, $P = 0.039$; AMY: open: $t_{18} = 3.08$, $P = 0.0065$: closed stop: $t_{18} = 1.62$, $P = 0.12$, two-sided Student's $t$-test). Taken together, these results demonstrate that VNS applied to stress-susceptible mice restored VN spike activity and 20–30 Hz LFP patterns in the PFC, which is consistent with the restoration of their behavioral patterns.

## Discussion

In this study, we demonstrated that VN activity is attenuated in stress-susceptible mice with altered anxiety behavior but not in stress-resilient mice. Simultaneous recordings of VN activity and PFC-AMY LFP signals revealed pronounced changes in 2–4 Hz and 20–30 Hz powers across environments with different behavioral states in the EPM test, which were disrupted in both vagotomized and stress-susceptible mice (Supplementary Fig. 4). Chronic treatment with VNS in stress-susceptible mice restored VN activity and behavior-relevant PFC and AMY LFP patterns.

Increased anxiety is a hallmark of stress-induced psychiatric and cognitive disorders. While EPM tests are widely used to assess anxiety levels in experimental rodent animals, behavioral changes (e.g., time in open/closed arms) induced by SD stress are not consistent among previous studies[30–35], probably because of the differences in experimental conditions. While our study detected no apparent stress-induced changes in open arm time in stress-susceptible mice, we

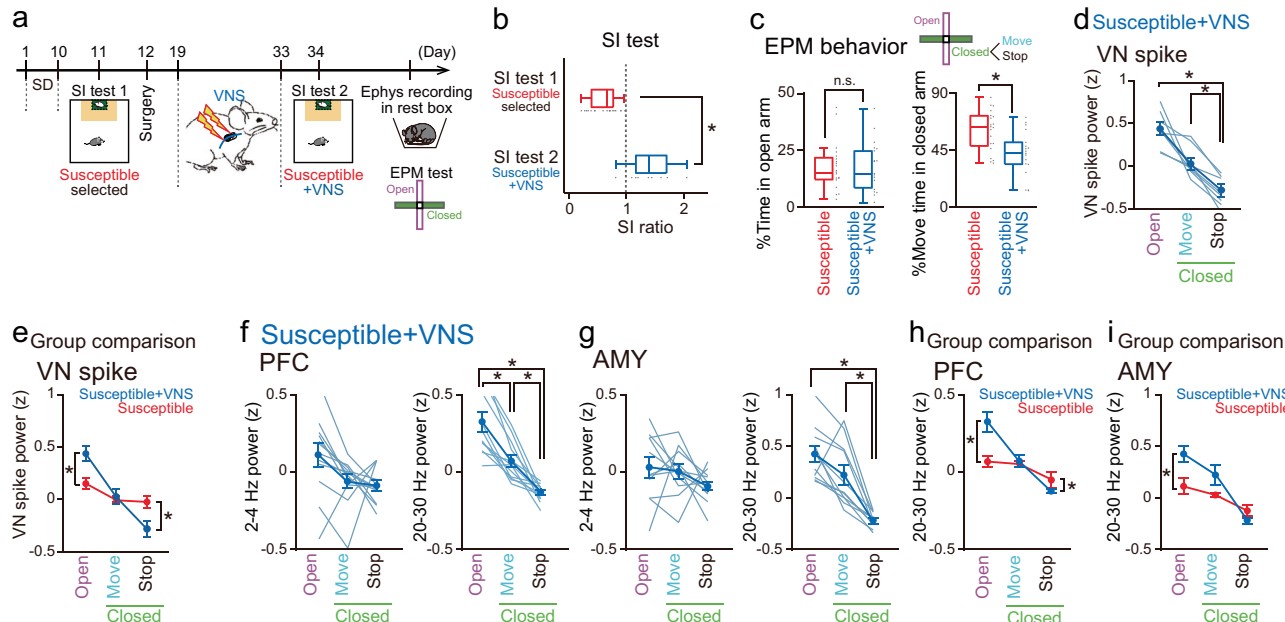

**Fig. 5 | VNS restores stress-induced PFC-AMY oscillations. a** VNS was applied to stress-susceptible mice. **b** SI ratios ($n$ = 18 and 10 mice). Box plots show center line as median, box limits as the 25th percentile and the 75th percentile, whiskers as minimum to maximum values that are not outliers. $Z$ = 4.18, *$P$ = 3.0 × 10$^{-5}$, two-sided Mann–Whiney $U$ test. **c** The percentage of the open arms (left; $t_{37}$ = 0.043, $P$ = 0.97, two-sided Student's $t$-test) and the move states in the closed arms (right; $t_{37}$ = 3.04, *$P$ = 0.0044, two-sided Student's $t$-test) ($n$ = 18 mice). Box plots show center line as median, box limits as the 25th percentile and the 75th percentile, whiskers as minimum to maximum values that are not outliers. **d** VN spike power in the susceptible + VNS group ($n$ = 7 mice). Data are presented as mean ± SEM. open vs closed stop, $t_6$ = 5.88, *$P$ = 0.0032; closed move vs closed stop, $t_6$ = 7.72, *$P$ = 7.4 × 10$^{-4}$; two-sided paired $t$-test followed by Bonferroni correction. **e** Across-group comparisons of VN spike power. open: $t_{19}$ = 4.19, *$P$ = 5.0 × 10$^{-4}$: closed stop: $t_{19}$ = 2.61, *$P$ = 0.017, two-sided Student's $t$-test. **f** PFC power ($n$ = 11 mice). Each thin line represents each mouse. Data are presented as mean ± SEM. 20–30 Hz: open vs closed move, $t_{10}$ = 5.12, *$P$ = 0.0013; open vs closed stop, $t_{10}$ = 6.09, *$P$ = 0.00035; closed move vs closed stop, $t_{10}$ = 4.76, *$P$ = 0.0023, two-sided paired $t$-test followed by Bonferroni correction. **g** Same as **f** but for AMY power ($n$ = 10 mice). 20–30 Hz: open vs closed stop, $t_9$ = 7.87, *$P$ = 7.6 × 10$^{-5}$; closed move vs closed stop, $t_9$ = 5.26, *$P$ = 0.0016; two-sided paired $t$-test followed by Bonferroni correction. **h, i** Across-group comparisons of PFC and AMY 20–30 Hz power. PFC: open: $t_{24}$ = 3.47, *$P$ = 0.020: closed stop: $t_{24}$ = 2.18, *$P$ = 0.039; AMY: open: $t_{18}$ = 3.08, *$P$ = 0.0065, two-sided Student's $t$-test. Source data are provided as a Source Data file.

detected a difference between resilient and susceptible phenotypes when focusing on the proportion of move states within closed arms, which has been used as a measure of anxiety[36]. Similar to the stress-susceptible mice, while vagotomized mice did not show apparent changes in their open arm time, in accordance with previous observations[3,6], they showed prominent increases in move states in the closed arms. The common behavioral changes in stress-susceptible and vagotomized mice imply that stress susceptibility is related to VN activity.

Consistent with this finding, our VN recordings demonstrated that the overall spike activity recorded from the bundle of the cervical VN was substantially reduced in stress-susceptible mice. These pathophysiological changes in VN activity could be explained by complex integration of reductions in signal transfers from the peripheral organs to the VN (i.e., the detectability of peripheral organ states by the VN) and/or reductions in peripheral organ functions that could originally activate the VN. In particular, the digestive system has dense connections with the VN and, thus, likely serves as a strong driver of VN activity. Indeed, the importance of the gut-brain communication through the VN in the modulation of mental states has been suggested by the observation that compositional changes in intestinal bacteria and external manipulation of intestinal cell activity exert anxiolytic and antidepressant effects[3–8]. Further studies are required to unveil the detailed physiological factors related to the VN by developing sophisticated techniques to finely record and manipulate the branches of the VN innervating individual peripheral organs and genetically identify subtypes of the VN.

We demonstrated that PFC 2–4 Hz power was negatively correlated with VN spike power and was reduced in open environments, which was disrupted in stress-susceptible and vagotomized mice. This frequency band is slightly lower than the PFC theta range (4–10 Hz) oscillations related to anxiety behavior[21–25] and the PFC 4–7 Hz oscillations related to social behavior[37]. A common feature of these PFC oscillations with frequency bands of <10 Hz is their power increases during increased anxiety, suggesting that these oscillations reported in different studies are expressed partly through common physiological mechanisms and sources. The involvement of PFC interneurons is suggested by the observation that the selective activation of PFC interneurons replicates 4-Hz oscillations[38,39]. While we did not identify cell types in the subset of PFC neurons that were phase-locked to the 2–4 Hz oscillations, most of these phase-locked cells had relatively high basal firing rates of >5 Hz and were thus likely composed of interneurons. On the other hand, PFC 20–30 Hz power was positively correlated with VN spike power and was elevated in open environments with increased VN spike power, which was also disrupted in stress-susceptible and vagotomized mice. These results suggest that PFC oscillations in this frequency band are an appropriate physiological substrate related to VN activity levels and that such vagal-brain interactions are helpful in mediating anxiety behavior.

VNS has been clinically applied to treat psychiatric diseases such as treatment-resistant depression[9,10,40]. Recently, a non-invasive transcutaneous auricular VNS method was developed[41–44]. Consistently, rodent studies have demonstrated that manipulation of VN activity can alter brain electrical activity[45–48] and evoke emotional and anxiolytic responses[14,15,49,50]. Our VN recordings confirmed that VNS substantially restored the VN activity that was weakened by SD stress, altered anxiety behavior, and dysfunctional PFC and AMY LFP dynamics. While our study showed that our VNS protocol with an interstimulus interval

of 1 min and a total duration of 3 h per day was effective, further studies are needed to conceive more appropriate parameters (e.g., duration and interval) of VNS, which may solve the problem of VNS patterns empirically utilized in both basic and clinical studies that vary considerably owing to their high degree of freedom in time.

Taken together, our study showed that stress-induced behavioral states and behavior-relevant PFC-AMY neuronal oscillations are crucially mediated by VN activity. Accumulation of these physiological insights into vagal-brain communications will help to identify the contributions of vagal signals to anxiety in pathological conditions such as depressive states and provide precise biological targets for VNS-based therapy for mood disorders.

## Methods

### Approvals
All experiments were performed with the approval of the animal experimental ethics committee at the University of Tokyo (approval number: P29-14) and the committee on animal experiments at Tohoku University (approval number: 2022 PhA-004) and in accordance with the NIH guidelines for the care and use of animals.

### Subjects
Male C57BL/6 J mice (eight to ten weeks old) with weighs of 29–35 g were subjected to social defeat (SD) stress, behavioral testing, and electrophysiological recordings. In addition, male CD-1 mice (more than 13 weeks old) with weights of 40–50 g were used as resident mice that imposed social defeat stress. All mice were purchased from SLC (Shizuoka, Japan).

They were maintained with free access to water and food under 12-h light/12-h dark schedule under housing conditions at 23 ± 1 °C with relative humidity of 50 ± 5% with lights off at 8:00 AM.

### SD stress
At least 1 week before beginning the social defeat experiment, all resident CD-1 mice more than 13 weeks of age were singly housed on one side of a home cage (termed the "resident area"; 42.5 cm × 26.6 cm × 15.5 cm). The cage was divided into two identical halves by a transparent Plexiglas partition (0.5 cm × 41.8 cm × 16.5 cm) with perforated holes, each with a diameter of 10 mm. The bedding in the resident area was left unchanged during the preoperative period. First, resident CD-1 male mice were screened for aggressor mice in SD experiments by introducing an intruder C57BL/6 J mouse that was specifically used for screening into the home cage during three 3-min sessions on 3 subsequent days. Each session included a different intruder mouse. CD-1 mice were selected as aggressors in subsequent experiments based on three criteria: during the three 3-min sessions, (1) the mouse attacked in at least two consecutive sessions, (2) the latency to initial aggression was less than 60 s, and (3) the above two criteria were met for at least two consecutive days out of three test days. Based on these criteria, only the mice screened as aggressor mice were utilized in the following SD stress paradigm.

To impose SD stress to an intruder mouse (C57BL/6 J mouse), an intruder mouse was introduced into the resident area including an aggressor mouse for a 5–10-min interaction. The interaction period was immediately terminated if the intruder mouse had a wound and bleeding due to the attack, resulting in interaction periods of 5–10 min. After the physical contact, the intruder mouse was transferred across the partition and placed in the opposite compartment of the home cage of the aggressor mouse for the following 24 h; this allowed the intruder mouse to have sensory contact with the aggressor mouse without physical contact. Over the following 10-day period, the intruder mouse was exposed to a different aggressor mouse so that the mice did not habituate the same residents.

### Preparation of an electrode assembly
The electrode assembly was composed of an electrical interface board (EIB) (EIB-36-PTB, Neuralynx, Inc., Bozeman, MT) that consisted of an outer cover and a core body created by 3D printers (UP Plus2). The EIB had a sequence of metal holes (channels) for connections with wire electrodes, including up to 24 brain LFP channels, 1 EMG channel, 1 VN channel, 1 reference channel for VN, and 2 ground/reference (g/r) channels. The individual channels from the peripheral areas were connected at the final step of the surgery.

### Surgery
C57BL/6 J mice were anesthetized with 3% isoflurane gas and then maintained with 1–3% isoflurane gas while lying on their backs. Veterinary ointment was placed on the mouse's eyes to prevent dryness. For all steps of an incision, the skin was sterilized with betadine and 70% ethanol. For each mouse, an incision was made in the left neck area from the larynx to the sternum, and the bundle including the VN and the carotid artery were isolated. The left cervical VN was isolated from the surrounding tissue and the left carotid artery. Here, we targeted the left VN because (1) the left VN has fewer connections to the sinoatrial node, compared with the right VN, and the left VNS thus induces fewer detrimental side effects on cardiac activity in both humans and rodents[51], and (2) the left VNS has been shown to be sufficient to induce anxiolytic effects and affect emotion-related functions[9,14–16,52]. For VN recordings, the isolated cervical VN was enclosed by a custom-made cuff-shaped VN electrode with the electrode contact sites attached to the inside walls of the cylindrical tube (inner diameter = 0.3 mm, electrode area = 0.15 mm$^2$, cathode–anode interval = 2.0 mm, and total length 4.0 mm). A reference electrode (stainless-steel wires; AS 633, Cooner Wire Company) with an electrical isolation was placed on the salivary gland that is located directly above (very close to) the cuff-shaped electrode implanted on the VN, which could minimize noise artifacts (such as myoelectric potential and minor physical fluctuation of electrode) around the recorded area. The open ends of these electrodes were extruded from the incision.

For vagotomy, the cervical VN bundle was exposed and dissected by surgical micro scissors. After implantation of the VN electrode, LFP electrodes were implanted into the brain. An electrode assembly that consisted of up to 7 tetrodes was stereotaxically implanted above the left PFC (2.0 mm anterior, 0 mm lateral and 1.8 mm depth to bregma) and the left AMY (1.5 mm posterior, 3.5 mm lateral and 4.5 mm depth to bregma). The tetrodes were constructed from 17-μm-wide polyimide-coated platinum-iridium (90/10%) wires (California Fine Wire), and the electrode tips were plated with platinum to lower the electrode impedances to 200–250 kΩ. In all surgeries, an additional incision was made at the incised neck area, and one EMG electrode (stainless-steel wires; AS 633, Cooner Wire Company) was sutured to the dorsal neck muscles. Ground electrodes were located on the cerebellum. The recording device was secured to the skull using stainless steel screws and dental cement. The open edges of the EMG electrodes were soldered to the EIB. The open edges of the VN electrodes were soldered to a socket attached to the electrode assembly. For all the mice, we implanted all electrodes for simultaneous electrophysiological recordings of VN spikes (except vagotomized mice), EMG signals, and PFC and AMY LFP signals. After all surgical procedures were completed, anesthesia was discontinued, and the mice were allowed to awaken spontaneously. Following surgery, each mouse was housed in a transparent Plexiglas cage with free access to water and food. For spike recordings, the tetrodes were advanced to the PFC over a period of at least one week following surgery.

### Electrophysiological recording and VNS
While all mice were implanted with electrodes for simultaneous electrophysiological recordings of VN spikes (except vagotomized mice), EMG signals, and PFC and AMY LFP signals, some electrodes were

implanted outside the target regions or accidentally disrupted during chronic implantations. In these cases, we recorded as many signals as possible simultaneously from the single mice. The mouse was connected to the recording equipment via Cereplex M (Blackrock Microsystems), a digitally programmable amplifier, which was placed close to the head. The output of the headstage was conducted to the Cereplex Direct recording system, a data acquisition system, via a light-weight multiwire tether and a commutator. LFP and EMG signals were sampled at 2 kHz and low-pass filtered at 500 Hz. The unit activity in LFP signals was amplified and bandpass filtered at 750 Hz to 6 kHz. Spike waveforms above a trigger threshold (50 µV) were time-stamped and recorded at 30 kHz in a time window of 1.6 ms. For VN signal recording, the socket was connected to open channels on the EIB through a wire. The sampling rate was 30 kHz. The moment-to-moment position was tracked at 15 Hz using a video camera attached to the ceiling. The frame rate of the movie was downsampled to 3 Hz, and the instantaneous speed of each frame was calculated based on the distance traveled within a frame (~333 ms). In the following analyses, video frames with massive optical noise or periods that were not precisely recorded due to temporal breaks of image data processing were excluded.

On recording days, the mice first rested in its home cage for at least 30 min (pre-rest), performed an EPM test for 10–20 min, and again rested in the same home cage (post-rest) for at least 30 min (Supplementary Fig. 1a).

To apply VNS to stress-susceptible mice, the socket was connected to an electrical stimulator through a wire. Stimulation (VNS; 0.8 mA, 0.1 ms pulse width; 20 Hz; 30 s duration) was performed every 1 min for 3 h in a day. These individual parameters were set to be in the same ranges to parameters used in previous studies[15,17,53,54].

### Social interaction (SI) test
SI tests were performed inside a dark room in a square-shaped box (39.3 cm × 39.3 cm) enclosed by walls 27 cm in height. A wire-mesh cage (6.5 cm × 10 cm × 24 cm) was centered against one wall of the arena during both no target and target sessions. Each SI test included two 150-s sessions (separated by an intersession interval of 30 s) without and with a target CD-1 mouse present in the mesh cage, termed no target and target sessions, respectively. In the no target session, a test C57BL/6 J mouse was placed in the box and allowed to freely explore the environment. The C57BL/6 J mouse was then removed from the box and a target CD-1 mouse was next introduced into the mesh cage. The design of the cage allowed the mouse to fit its snout and paws through the mesh cage but not to escape from the cage. In the target session, the same C57BL/6 J mouse was placed beside the wall opposite to the mesh cage. In each session, the time spent in the interaction zone, a 14.5 cm × 24 cm rectangular area extending 8 cm around the mesh cage was quantified. A social interaction (SI) ratio was computed as the ratio of time spent in the interaction zone in the target session to the time spent there in the no target session.

### Elevated plus maze (EPM) test
An EPM was made of ABS resin and consisted of a central square (7.6 × 7.6 cm) and four arms (28 cm long × 7.6 cm wide, two open arms with no railing and two closed arms enclosed by a transverse wall 15 cm in height). The maze was elevated 30 cm from the floor. In a recording session, a mouse was placed in the center of the central square facing the open arm and allowed to explore the maze apparatus for 10–20 min. In behavioral analyses, the first 10 min of the EPM test was analyzed in all mice. Time in which the mice stayed in the open and closed arms was calculated. The frame rate of the movie was downsampled to 3 Hz, and the instantaneous speed of each frame was calculated based on the distance traveled within a frame (~333 ms).

### Histological analysis to confirm electrode placement
The mice were overdosed with isoflurane, perfused intracardially with 4% paraformaldehyde in phosphate-buffered saline (PBS; pH 7.4) and decapitated. After dissection, the brains were fixed overnight in 4% paraformaldehyde and equilibrated with 30% sucrose in PBS for an overnight each. Frozen coronal sections (50 µm) were cut using a microtome, and serial sections were mounted and processed for cresyl violet staining. For cresyl violet staining, the slices were rinsed in water, stained with cresyl violet, and coverslipped with Permount. The positions of all electrodes were confirmed by identifying the corresponding electrode tracks in histological tissue.

### Immunostaining
The slices were rinsed with 1% bovine serum albumin (BSA) in PBS for 10 min three times, permeabilized with 0.1% Triton X-100 in PBS for 10 min, and then incubated with 1% BSA in PBS for 60 min. The slices were then incubated with a primary mouse anti-c-Fos antibody (EnCor Biotechnology, Cat# MCA-2H2, 1:1000) and 1% BSA in PBS for one overnight period at 4 °C. After rinsing with 1% BSA in PBS for 10 min three times, the slices were then labeled with a secondary anti-mouse IgG antibody Alexa 488 (Invitrogen, Cat# A-11029, 1:500) and 1% BSA in PBS for one overnight period at 4 °C. After rinsing with 1% BSA in PBS for 10 min three times, the slices were enclosed. Images were acquired using a fluorescent microscope (BZ-X800; Keyence) with an objective lens (×10, 0.4 NA; ×20, 0.75 NA).

### EMG analysis
In rest periods, movement/quiescent periods were defined based on EMG signals. EMG signals were band-pass filtered at 20–200 Hz, and root-mean-square (RMS) values were calculated from the filtered EMG signals with a bin size of 1 s. As baseline levels of movement were considerably variable across mice, fixing a certain EMG threshold to define movement in all mice was practically impossible. We thus manually adjusted EMG thresholds for individual mice by scrutinizing their EMG RMS traces. In the majority of mice, the thresholds of EMG signals to define movement/quiescent periods were set to approximately 8 standard deviations (SDs) above the mean of baseline EMG RMS traces.

### VN spike analysis
To reduce background electrical noise and increase the signal to noise ratio[55–57], an electrical signal from a VN electrode was subtracted from that from a reference electrode placed on the salivary gland, termed a VN signal. During the movement periods defined in the rest periods and during an EPM test, isolation of compound VN traces into individual spike-like signals was impossible. We thus collectively computed these signals by band-pass filtering VN signals at 300–1000 Hz and computing root-mean-square (RMS) values from the filtered VN signals with a bin size of 1/3 s. During the quiescent periods defined in the rest periods, VN signals were composed of compound spikes, but their single spike-like signals were more clearly visible (magnified in Fig. 1f). VN signals from a quiescent period of more than 5 min were manually extracted based on EMG RMS traces. During these periods, a spike unit was detected when the amplitude of a negative deflection of the filtered VN signals exceeded a threshold during a period with EMG RMS exceeding a threshold as described above. Similar to EMG signals, baseline levels of filtered VN signals were considerably variable across mice. We thus manually adjusted the detection thresholds for individual mice by scrutinizing their filtered VN signals. The thresholds were set to approximately 5 standard deviations (SDs) below the mean of baseline filtered VN signals. Burst spike units detected within a 5-ms bin were regarded as a single spike. In some mice, these detected spike units contained spike-like signals that were perfectly locked to heartbeat with a constant interval of 110–130 ms. In that case, these signals

were excluded by manually setting a threshold to define heartbeat timings in VN signals band-pass filtered at 10–200 Hz.

## LFP analysis

To compute the time-frequency representation of LFP power, brain LFP signals were convolved using complex Morlet wavelet transformation by the Matlab. The signals that included apparent electrical noise due to physical striking of the animal's head to the walls were manually removed. LFP power in each frequency band was zscored based on the average and SD of LFP power at the frequency band in an entire recording period. When data were obtained from multiple electrodes in a mouse, an electrode was selected so that each mouse had a single value.

## Spike unit analysis

The data comprised spike patterns of 47 PFC neurons recorded with tetrodes from 6 mice during resting in a home cage with a size of $25 \times 20$ cm. Spike sorting was performed offline using the graphical cluster-cutting software Mclust. Clustering was performed manually in 2D projections of the multidimensional parameter space (i.e., comparisons between the waveform amplitudes, the peak-to-trough amplitude differences, the waveform energies, and the first principal component coefficient (PC1) of the energy-normalized waveform, each measured on the four channels of each tetrode). Only clusters that could be stably tracked across recording periods were considered to be the same cells and were included in our analysis. For each cell, the degree of phase locking during a rest period was analyzed. For each cell, the phase-spike rate distribution was computed by plotting the firing rate as a function of the phase of 2–4 Hz, 20–30 Hz LFP traces, divided into bins of 30° and smoothed with a Gaussian kernel filer with standard deviation of one bin (30°), and a Rayleigh $r$-value was calculated as mean vector length (MVL). To evaluate a MVL of a cell, we created shuffled datasets in which spike timing was randomized within a recording period and MVL was similarly computed from 1000 shuffled datasets, termed MVL$_{shuffled}$. The MVL of an original data was considered to be significant ($P < 0.001$) when the MVL was higher than the top 0.1% of the corresponding MVL$_{shuffled}$.

## Statistical analysis

All data were presented as the mean ± standard error of the mean (SEM), unless otherwise specified, and were analyzed using MATLAB. For comparisons across mouse groups, data are displayed as boxplots and analyzed by Mann-Whitney $U$ test. For comparisons of c-fos expressions between mouse groups, datasets were resampled as many times as the number of samples by a bootstrap method in each group. The bootstrapped datasets in each resampling were averaged, and Student's $t$-test were applied. For comparisons of physiological measures within single mice, data points are displayed in addition to sample mean and SEM and analyzed by paired $t$-test after confirming data normality by the $F$ test. Multiple group comparisons were performed by post hoc Bonferroni corrections. The null hypothesis was rejected at the $P < 0.05$ level.

## Reporting summary

Further information on research design is available in the Nature Portfolio Reporting Summary linked to this article.

## Data availability

Original physitological datasets are provided on Mendeley Data (https://data.mendeley.com/datasets/6yy34xkfmb/1). Source data are provided with this paper.

## Code availability

Original codes are provided on Mendeley Data (https://data.mendeley.com/datasets/6yy34xkfmb/1).

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

## Acknowledgements

This work was supported by KAKENHI (19H04897; 20H03545; 21H05243) from the Japan Society for the Promotion of Science (JSPS), grants (JP21zf0127004) from the Japan Agency for Medical Research and Development (AMED), grants from the Japan Science and Technology Agency (JST) (JPMJCR21P1; JPMJMS2292), Lotte Research Promotion Grant, the Uehara Memorial Foundation, and Research Foundation for Opto-Science and Technology to T. Sasaki; grants from the JST Exploratory Research for Advanced Technology (JPMJER1801), and Institute for AI and Beyond of the University of Tokyo to Y. Ikegaya; and a JSPS Research Fellowship to T. Okonogi, N. Kuga, and T. Kayama.

## Author contributions

T.O. and T.S. designed the study. T.O., M.Y., N.K., and T.K. acquired the electrophysiological data and the behavioral data. M.Y. performed immunostaining. T.O. and T.S. prepared the figures. Y.I. supervised the project. T.S. wrote the main paper text. All the authors reviewed the paper text.

## Competing interests

The authors declare no competing interests.
