## [Peer Review File · Nature Communications]

Stress-induced vagal activity influences anxiety-relevant prefrontal and amygdala neuronal oscillations in male mice
TitleREVIEWER COMMENTS

Reviewer #1 (Remarks to the Author):

Vagus nerve stimulation (VNS) has been shown to ameliorate psychopathology such as anxiety. In this study, the authors used a chronic social defeat stress model to examine the effect of VNS on brain oscillation during anxiety related behaviors. They found that vagus nerve activity of mice can be increased in mice when they are on an elevated plus maze (EPM). Anxiety related behaviors on the elevated plus maze can modulate the 2-4 Hz and 20-30 Hz oscillations in the mPFC and amygdala, respectively. In mice that were susceptible to chronic social defeat stress, anxiety-induced modulations of mPFC and amygdala activities were gone. VNS not only can modulate the oscillations of mPFC and amygdala, it also restored these oscillation in stress susceptible mice. They concluded the vagal brain relationship could underlie the VNS-based therapy for mood disorders.

Reduced vagal activity has been reported after chronic stress and social defeat. VNS has been used for treating treatment resistant mood disorders. Brain imaging studies have revealed changes in activities of the frontal lobe and the amygdala during VNS. While the modulation of mPFC and amygdala activity after VNS in the current study is not too surprising, the examination of the effect of VNS on mPFC and amygdala oscillations during anxiety related behaviors is quite novel. The fact that VNS could modulate these oscillations during EPM and to rescue the deficit of these oscillations in stress susceptible mice is highly interesting. While many studies inferred VN activity indirectly through cardiovascular functions, extracting VN activity from electrophysiological recording in the current study is a notable advancement.

While I appreciate the effort of the team to relate mPFC and AMY oscillations with anxiety behaviors, my first concern is the unclear relationship between move time in closed arm and anxiety. In the quoted paper (ref 39) that examined Move vs Stop behaviors in closed arm, stop behaviors during closed were associated with changes in mPFC oscillations. This association, however, added little information about the anxiety state of the mice. Comparing between resilient and susceptible mice, no difference in VN spike power was found between Move and Stop states. Although no change in the open arm time was found between all animal groups, the VN power of resilient mice in open arm was higher than that in both closed arm states. It seems that resilient mice but not susceptible mice were able to produce VN spikes during the anxious state while the animals were in open arm. Perhaps changes in brain oscillations during open arm could be more relevant to the anxiety state of tested mice.

Another concern is the lack of data from resilient mice. Although PFC and AMY LFP power during EPM in susceptible mice are different from stress naïve mice, no data of PFC and AMY LFP recording from resilient mice were shown. It is unclear if changes in LFP power is not a general effect of chronic stress. Showing normal PFC and AMY LFP during EPM in resilient mice will address this concern.

Other points

- Line 34: What is the rationale that VN activity could encodes the anxiety states of individuals? Could changes in VN activity represent a feedback mechanism of mice to counter the expression of anxiety behaviors?
- Line 47: It seems counterintuitive to say that VNS could 'restore' anxiety behavior.
- Since LFP and VN recording were done at various time points after social defeat stress, it is important to show that the susceptible phenotype (social avoidance in the SI test) remains stable for weeks after stress. For instance, it is unclear to me if the effect of VNS on social avoidance is an antidepressant effect or the weakening of this phenotype a few weeks after stress.
- In Fig. 1F, susceptible mice showed lower spike rate than control mice. However, all other comparisons of VN activity were based on changes in VN spike power. Please provide data to compare VN spike power between the 3 animal groups.

- Since the significance of spike phase locking was defined as $p < 0.05$ from shuffled datasets, the 6.4% phase locked PFC neurons for the 20-30 Hz oscillations seem to be too close to the chance level.
- What was the effect of chronic VNS on basal levels of VN activity?
- Apart from regulating anxiety related behaviors, the mPFC regulates social behavior. It would be interesting to find out if changes in mPFC oscillation after VNS in susceptible mice is also related to the rescue of social impairment in susceptible mice.
- No mentioning of sex information of mice was provided. Since most studies of social defeat used only male mice, I assumed only male mice were used in the study. It is however important to clarify that in the methods.

Reviewer #2 (Remarks to the Author):

Although VNS is used as a therapy for depression and it is being tested for use as a treatment for other mood disorders, the mechanisms of action are largely unknown. This study used simultaneous recordings of vagus nerve activity and local field potentials (LFPs) in the amygdala and prefrontal cortex in naive, stressed, and vagotomized mice. Mice were exposed to resident intruder stress and then given a social interaction test. Mice showing a preference for social interaction were considered resilient and mice that avoided another mouse during social interaction testing were considered susceptible. Vagus nerve activity during a quiescent period was diminished in stress-susceptible mice when compared to vagus nerve activity in naive mice. No differences were observed in time spent in open arms of an elevated plus maze, but susceptible mice spent more time moving in the closed arms than naive mice. Vagus nerve activity increased in resilient and naive mice during exploration of open arms, but no change was observed in stress-susceptible mice. Interestingly, differences in LFP power were observed in naive and resilient mice while on the elevated plus maze, but these differences were not seen in stress-susceptible mice. Vagotomized mice performed like stress-susceptible mice in the elevated plus maze and no change was observed in prefrontal cortex or amygdala LFPs in open vs. closed arms of the elevated plus maze. Finally, vagus nerve stimulation reversed social interaction deficits and restored normal LFP patterns in susceptible mice. This work is important because it may shed light on anxiety-related circuitry and identify targets and methods for treatment of mood disorders. Original datasets will be provided on Mendeley Data.

Limitations of the study are:

1. I think the title could be more accurate and informative. Something like "Stress-induced vagal activity influences anxiety-relevant prefrontal and amygdala neuronal oscillations". There was no real analysis of coherence or what leads what in studies of PFC and amygdala oscillations. Stress-susceptible vagal activity is confusing because stress-susceptible mice show less vagal activity.
2. Some copy editing is needed to correct grammatical errors.
3. Some statements are too strong. It remains to be determined whether differences in locomotion in the closed arms of the EPM is an indication of altered anxiety. Also, lines 32-35 suggest that the vagal signaling is the fundamental basis for emotional states and this encodes anxiety states. It would be safer to say that the vagus is part of a system that sustains emotional states, and it signals anxiety states, rather than encodes them. Lines 507-509 indicate that PFC oscillations are the substrate for VN afferents to mediate anxiety behavior, but there was no direct test of causality and cervical vagal activity may have effects beyond those observed here. For example, cervical vagal activity would also presumably correlate with activity of descending vagal fibers which have been implicated in anxiolytic effects of VNS (see Noble et al., 2019).
4. It is not clear whether both male and female mice are used in these experiments. The SAGER guidelines should be considered and an explanation provided if only one sex is used.
5. Based on the explanation of selection of aggressors, it seems like the experimental mice are exposed only to mice previously characterized as "aggressors", but please confirm. Also, the sex of the aggressors should be included.
6. The description of the electrode assembly is helpful and the picture is appreciated, but it is so small that it is impossible to make out even at 200% of original size. Likewise, the image of cFos in the PFC

and amygdala is too small to evaluate.

7. There is no explanation for why the LEFT vagus, PFC, and amygdala were targeted.

8. There is no explanation for why the salivary gland was used for the reference electrode.

9. There is no explanation for why the VNS parameters were selected.

10. Testing on the elevated plus maze seems to be given over a range of 10-20 min. It is not clear if some mice explored for 10 min while others explored for 20 min, and whether they were compared across conditions. This is important because behavior on the maze changes over time.

11. Interesting vagus nerve spike data were acquired during the rest period, which preceded time on the elevated plus maze. It is understandable that the rest periods provided better conditions for recording, but it may be even more interesting to measure vagus nerve activity after testing on the elevated plus maze, based on research by James McGaugh and Cedric Williams indicating that adrenaline exerts its effects on the brain AFTER exposure to stressful events, via vagus nerve signaling.

12. It is interesting that there are no vagus nerve activity differences between susceptible and resilient mice. Likewise, PFC and amygdala activity were not compared across susceptible and resilient mice. This makes one wonder whether exposure to social defeat, susceptibility to stress, or the combination mediates these effects.

Overall, this is a cohesive set of experiments representing work that will be of significance to the field and related fields.

Reviewer #3 (Remarks to the Author):

This is an interesting study investigating the effects of stress on vagal induced oscillations in the brain

The subdivision of animals into susceptible & resilient needs to be justified as many animals will actually be in the middle. Wouldn't extremes be better to determine stress sensitivity.

The Elevated plus maze is not the best task to assess amygdala function. Fear conditioning based assays would be better.

The fact that stress does not alter the main parameters in the EPM is also problematic

The number of animals used is small throughout and would need justification

The use of a single sex needs justification

Manuscript number: NCOMMS-23-03900

"Stress-induced vagal activity influences anxiety-relevant prefrontal and amygdala neuronal oscillations"

Reviewer #1:

Overall comments:

Vagus nerve stimulation (VNS) has been shown to ameliorate psychopathology such as anxiety. In this study, the authors used a chronic social defeat stress model to examine the effect of VNS on brain oscillation during anxiety related behaviors. They found that vagus nerve activity of mice can be increased in mice when they are on an elevated plus maze (EPM). Anxiety related behaviors on the elevated plus maze can modulate the 2-4 Hz and 20-30 Hz oscillations in the mPFC and amygdala, respectively. In mice that were susceptible to chronic social defeat stress, anxiety-induced modulations of mPFC and amygdala activities were gone. VNS not only can modulate the oscillations of mPFC and amygdala, it also restored these oscillation in stress susceptible mice. They concluded the vagal brain relationship could underlie the VNS-based therapy for mood disorders.

Reduced vagal activity has been reported after chronic stress and social defeat. VNS has been used for treating treatment resistant mood disorders. Brain imaging studies have revealed changes in activities of the frontal lobe and the amygdala during VNS. While the modulation of mPFC and amygdala activity after VNS in the current study is not too surprising, the examination of the effect of VNS on mPFC and amygdala oscillations during anxiety related behaviors is quite novel. The fact that VNS could modulate these oscillations during EPM and to rescue the deficit of these oscillations in stress susceptible mice is highly interesting. While many studies inferred VN activity indirectly through cardiovascular functions, extracting VN activity from electrophysiological recording in the current study is a notable advancement.

We sincerely thank the reviewer for their thorough review of our manuscript and for recognizing the significance of our work. We have carefully addressed the comments and suggestions. Thanks to the comments, we believe the quality of this manuscript has been significantly enhanced.

Major points:

(1-1) While I appreciate the effort of the team to relate mPFC and AMY oscillations with anxiety behaviors, my first concern is the unclear relationship between move time in closed arm and anxiety. In the quoted paper (ref 39) that examined Move vs Stop behaviors in closed arm, stop behaviors during closed were associated with changes in mPFC oscillations. This association, however, added little information about the anxiety state of the mice.

(Answer)

Thank you for this crucial comment. As the reviewer suggested, the relationship between

move time in closed arms and anxiety has not been reported. To tell the truth, when we first designed this study, we expected to see changes in open arm time. Unfortunately, we could not detect significant reductions in open arm time by stress. As described in the Discussion part (Line 345-347), the effects of SD stress on time in open/closed arms are not consistent among previous studies.

We instead noticed that stress-susceptible mice apparently showed unstable behavior when they entered into closed arms. We therefore defined and quantified the proportion of move states within closed arms. Interestingly, these behavioral changes were also observed from vagotomized mice and could be restored by VNS. While we cannot discuss the exact mechanisms underlying this behavior, we would like to report these consistent results as a new measure of anxiety-related behavior supported by vagus nerve activity and LFP oscillations found in this study. These are discussed in Line 348-355.

(1-2) Comparing between resilient and susceptible mice, no difference in VN spike power was found between Move and Stop states. Although no change in the open arm time was found between all animal groups, the VN power of resilient mice in open arm was higher than that in both closed arm states. It seems that resilient mice but not susceptible mice were able to produce VN spikes during the anxious state while the animals were in open arm. Perhaps changes in brain oscillations during open arm could be more relevant to the anxiety state of tested mice.

(Answer)

Thank you for this crucial suggestion. This point was also raised by the other reviewer (Reviewer 2 Comment #12). To address the other comments, we first increased the number of mice, especially for resilient mice (Figure 1f (VN), 1g, 1h (behavior), 1i, 1j (VN), 3a, 3b, 3e, and 3f (PFC and AMY)). The new datasets demonstrated that resilient mice, similar to naïve mice, showed significantly larger VN spike power and higher VN spike rates than stress-susceptible mice. These results demonstrate that intrinsic VN activity during quiescent periods was reduced in stress-susceptible mice, but not in stress-resilient mice (described in Line 92-103).

With these results, we consider that this reviewer's concern has been alleviated.

Furthermore, Figure 3a and 3b demonstrate that, in resilient mice, 20–30 Hz LFP power in the PFC and AMY in the open arms and during move periods in the closed arms was significantly higher than that during stop periods in the closed arms, as observed from naïve mice. These results demonstrate that anxiety-related LFP power changes in the PFC-AMY circuits were specifically disrupted in stress-susceptible mice, but not in naïve and stress-resilient mice, consistent with the change patterns of VN power changes in the EPM test. These are described in Line 217-252.

(2) Another concern is the lack of data from resilient mice. Although PFC and AMY LFP power during EPM in susceptible mice are different from stress naïve mice, no data of PFC and AMY LFP recording from resilient mice were shown. It is unclear if changes in LFP power is not a general effect of chronic stress. Showing normal PFC and AMY LFP during EPM in resilient mice will address this concern.

(Answer)

Thank you for this crucial suggestion. We performed additional experiments from stress-resilient mice and presented data in Figure 1f (VN), 1g, 1h (behavior), 1i, 1j (VN), 3a, 3b, 3e, and 3f (PFC and AMY), similar to naïve and stress-susceptible mice.

The main findings from stress-resilient mice are as follows:

- Resilient mice showed larger VN spike power and higher VN spike rates than stress-susceptible mice (Fig. 1f).
- Resilient mice showed no significant differences in the duration of move states in the closed arms, compared with naïve mice (Fig. 1h).
- In resilient mice, VN spike power in the open arms was significantly higher than in the closed arms and VN spike power during move states in the closed arms was significantly higher than during stop states in the closed arms (Fig. 1i), similar to naïve mice.
- In resilient mice, 20–30 Hz LFP power in the PFC and AMY in the open arms and during move periods in the closed arms was significantly higher than that during stop periods in the closed arms (Fig. 3a and 3b)

Taken together, these results demonstrate that resilient mice showed similar anxiety-related behavior and behavior-relevant VN activity and PFC and AMY LFP patterns to those observed from naïve mice, which were distinct from stress-susceptible mice.

Other points

(1) Line 34: What is the rationale that VN activity could encodes the anxiety states of individuals? Could changes in VN activity represent a feedback mechanism of mice to counter the expression of anxiety behaviors?

(Answer)

Thank you for this crucial suggestion. As the reviewer indicated, there is no sufficient rationale to support “VN activity encodes anxiety”. We therefore rewrote our manuscript as

follows: “it remains unclear how VN activity undergoes dynamic changes in relation to the anxiety states of individuals and how such relationship between VN activity and anxiety are pathologically altered in mental disorders.” (Line 35-37)

(2) Line 47: It seems counterintuitive to say that VNS could ‘restore’ anxiety behavior.

(Answer)

As the reviewer suggested, this statement was “counterintuitive” in the Introduction part. We here replaced “restore” to “influence”. (Line 48)

(3) Since LFP and VN recording were done at various time points after social defeat stress, it is important to show that the susceptible phenotype (social avoidance in the SI test) remains stable for weeks after stress. For instance, it is unclear to me if the effect of VNS on social avoidance is an antidepressant effect or the weakening of this phenotype a few weeks after stress.

(Answer)

Thank you for asking this issue. We actually had a dataset to answer this comment (which has not been published anywhere).

In Supplementary Figure 1, we now presented datasets in which mice that were classified as stress susceptible and resilient phenotypes on the next day after 10-days SD stress were again tested on a SI test three weeks later. Supplementary Figure 1b demonstrates that there is a significantly positive correlation of SI ratios between one day (horizontal axis) and three weeks (vertical axis) after 10-days SD stress. In other words, the majority of the mice that were identified as stress-susceptible and stress-resilient phenotypes on the next day after SD stress continued to show the same phenotypes three weeks later. The long-lasting effect allowed us to monitor stress-related electrophysiological activity for at least three weeks. These results are described in Line 60-64.

Based on these results, we stated that the observed effect by VNS (Figure 5) is not simply explained by the spontaneous recovery of behavioral phenotypes for three weeks. We described this point in Line 294-298.

(4) In Fig. 1F, susceptible mice showed lower spike rate than control mice. However, all other comparisons of VN activity were based on changes in VN spike power. Please provide data to compare VN spike power between the 3 animal groups.

(Answer)

We now presented VN spike power (Fig. 1f, top) in parallel with VN spike rates (Fig. 1F,

bottom), showing similar statistical results in both of the two parameters.

(5) Since the significance of spike phase locking was defined as $p < 0.05$ from shuffled datasets, the 6.4% phase locked PFC neurons for the 20-30 Hz oscillations seem to be too close to the chance level.

(Answer)

Thank you for this crucial comment. Actually, the previous criterion ($P < 0.05$) was not stringent (Fig. 2h).

In the revised manuscript, we set much more strict criterion in which an original MVL was considered to be significant ($P < 0.001$) when the MVL was higher than the top 0.1% of the corresponding MVL_{shuffled} . This is now described in the Methods (Line 620-622).

With this new criterion ($P < 0.001$), 1 (out of 7) and 1 (out of 3) cells were excluded as non-significant cells in the 2–4 Hz and 20–30 Hz bands, respectively, compared from the previous version (with $P < 0.05$).

After all, 6 (12.8%) and 2 (4.3%) neurons out of the all recorded neurons (47 neurons) met this criterion and were identified as significant neurons. These probabilities of observing the significant neurons under the new strict criterion ($P < 0.001$) are not explained by the chance level and they are considered as true phase-locked neurons. These results are now described in the Results part (Line 191-193) and Figure 2h.

(6) What was the effect of chronic VNS on basal levels of VN activity?

(Answer)

To our understanding, this comment suggests that we present data of VN spike recordings during rest periods (as in Figure 1f), in addition to those from the EPM test (Figure 5d and 5e).

While we first tried to address this comment as much as possible, regrettably, we are sorry that we could not obtain enough datasets due to various technical problems such as electrode damage and inadequate recording conditions. As this experiment takes more than one month for one mouse (as shown in Figure 5a) and the success rates of VN recordings after chronic VNS are not high (less than 40%), it would take additional six or more months to completely address this comment.

We would greatly appreciate your understanding regarding our decision to forgo presenting the data.

(7) Apart from regulating anxiety related behaviors, the mPFC regulates social behavior. It would be interesting to find out if changes in mPFC oscillation after VNS in susceptible mice is also related to the rescue of social impairment in susceptible mice.

(Answer)

If we could properly understand this comment, we have been presenting datasets to answer this issue. The susceptible+VNS mouse group that showed the restoration of social behavior (as confirmed by SI ratios in Figure 5b) actually exhibited the restoration of anxiety-related behavior (Fig. 5c), anxiety-relevant VN spike activity (Fig. 5d and 5e), and anxiety-relevant PFC and AMY 20–30 Hz patterns (Fig. 5f-5i), which are comparable to those observed from the naïve mice. These results are described in Line 315-321.

(8) No mentioning of sex information of mice was provided. Since most studies of social defeat used only male mice, I assumed only male mice were used in the study. It is however important to clarify that in the methods.

(Answer)

As the reviewer indicated, only male mice were used in this study.

- In the Abstract, we added “male mice” (Line 12).
- In the Materials and Methods, we stated “male mice” at the section of “Ethical approval and animals”.
- In the Nature Portfolio Reporting summary, we described reasons why only male mice were included in this study as follows: “Only male mice were used in this study because the experimental paradigms of social defeat stress employed in this study has been established for male mice. Importantly, the stress models using male aggressor mice are only applicable to male mice.”.

Reviewer #2:

Overall comments:

Although VNS is used as a therapy for depression and it is being tested for use as a treatment for other mood disorders, the mechanisms of action are largely unknown. This study used simultaneous recordings of vagus nerve activity and local field potentials (LFPs) in the amygdala and prefrontal cortex in naive, stressed, and vagotomized mice. Mice were exposed to resident intruder stress and then given a social interaction test. Mice showing a preference for social interaction were considered resilient and mice that avoided another mouse during social interaction testing were considered susceptible. Vagus nerve activity during a quiescent period was diminished in stress-susceptible mice when compared to vagus nerve activity in naive mice. No differences were observed in time spent in open arms of an elevated plus maze, but susceptible mice spent more time moving in the closed arms than naive mice. Vagus nerve activity increased in resilient and naive mice during exploration of open arms, but no change was observed in stress-susceptible mice. Interestingly, differences in LFP power were observed in naive and resilient mice while on the elevated plus maze, but these differences were not seen in stress-susceptible mice. Vagotomized mice performed like stress-susceptible mice in the elevated plus maze and no change was observed in prefrontal cortex or amygdala LFPs in open vs. closed arms of the elevated plus maze. Finally, vagus nerve stimulation reversed social interaction deficits and restored normal LFP patterns in susceptible mice. This work is important because it may shed light on anxiety-related circuitry and identify targets and methods for treatment of mood disorders. Original datasets will be provided on Mendeley Data.

We sincerely thank the reviewer for their thorough review of our manuscript and for recognizing the significance of our work. We have carefully addressed the comments and suggestions. Thanks to the comments, we believe the quality of this manuscript has been significantly enhanced.

Limitations of the study are:

1. I think the title could be more accurate and informative. Something like "Stress-induced vagal activity influences anxiety-relevant prefrontal and amygdala neuronal oscillations". There was no real analysis of coherence or what leads what in studies of PFC and amygdala oscillations. Stress-susceptible vagal activity is confusing because stress-susceptible mice show less vagal activity.

(Answer)

Thank you for this crucial comment. As the reviewer suggested, "stress-susceptible vagal activity" in the title was not appropriate expression. We now employed the title suggested by the reviewer.

2. Some copy editing is needed to correct grammatical errors.

(Answer)

This manuscript was proofread by two experts in the revision process.

3-1. Some statements are too strong. It remains to be determined whether differences in locomotion in the closed arms of the EPM is an indication of altered anxiety.

(Answer)

Throughout the manuscript, we removed strong statements that the altered locomotion represents anxiety, weakened our expressions (e.g. “might represent”), or simply stated the fact (e.g. “altered behavioral states in the EPM test”). For details, please confirm the manuscript highlighted in blue in the Abstract (Line 18), the Results part (Line 129-132, Line 312-314), and the Discussion part (Line 391-392).

3-2. Also, lines 32-35 suggest that the vagal signaling is the fundamental basis for emotional states and this encodes anxiety states. It would be safer to say that the vagus is part of a system that sustains emotional states, and it signals anxiety states, rather than encodes them.

(Answer)

As the reviewer suggested, the term “encode” was a too strong statement. Here, we simply stated our scientific questions as follows: “it remains unclear how VN activity undergoes dynamic changes in relation to the anxiety states of individuals and how such relationship between VN activity and anxiety are pathologically altered in mental disorders.” (Line 35-37)

3.3 Lines 507-509 indicate that PFC oscillations are the substrate for VN afferents to mediate anxiety behavior, but there was no direct test of causality and cervical vagal activity may have effects beyond those observed here. For example, cervical vagal activity would also presumably correlate with activity of descending vagal fibers which have been implicated in anxiolytic effects of VNS (see Noble et al., 2019).

(Answer)

Similar to the Comment #3-2, this statement including “afferent” and “encoding” was too speculative. We now rewrote the sentence as follows: “These results suggest that PFC oscillations in this frequency band are an appropriate physiological substrate related to VN activity levels and such vagal-brain interactions are helpful to mediate anxiety behavior.” (Line 384-386)

4. It is not clear whether both male and female mice are used in these experiments. The SAGER guidelines should be considered and an explanation provided if only one sex is used.

(Answer)

Only male mice were used in this study.

- In the Abstract, we added “male mice” (Line 12).
- In the Materials and Methods, we stated “male mice” at the section of “Ethical approval and animals”.
- In the Nature Portfolio Reporting summary, we described reasons why only male mice were included in this study as follows: “Only male mice were used in this study because the experimental paradigms of social defeat stress employed in this study has been established for male mice. Importantly, the stress models using male aggressor mice are only applicable to male mice.”.

5. Based on the explanation of selection of aggressors, it seems like the experimental mice are exposed only to mice previously characterized as "aggressors", but please confirm. Also, the sex of the aggressors should be included.

(Answer)

Yes. We added more clear explanations for this protocol in the Methods and Results part as follows:

“First, resident CD-1 male mice were screened for aggressor mice in SD experiments... Based on these criteria, only the mice screened as aggressor mice were utilized in the following SD stress paradigm.

To impose SD stress to an intruder mouse (C57BL/6J mouse), an intruder mouse was introduced into the resident area including an aggressor mouse for a 5–10-min interaction...

Over the following 10-day period, the intruder mouse was exposed to a different aggressor mouse so that the animals did not habituate the same residents.” (Line 425-443)

“C57BL/6J male mice were subjected to 10-min SD stress from CD-1 male mice that were screened as aggressor mice for 10 days” (Line 55-56)

6. The description of the electrode assembly is helpful and the picture is appreciated, but it is so small that it is impossible to make out even at 200% of original size. Likewise, the image of cFos in the PFC and amygdala is too small to evaluate.

(Answer)

In Figure 1d, we presented larger pictures of cuff-electrodes with higher spatial resolutions and with their brightness and contrast adjusted.

In Figure 2a, we replaced images of c-fos expressions with larger pictures with higher spatial resolutions and with their brightness and contrast adjusted. In addition, typical images from control (sham) experiments are presented for comparisons.

7. There is no explanation for why the LEFT vagus, PFC, and amygdala were targeted.

(Answer)

The reasons why we selected the LEFT vagus nerve were as follows: (1) the left VN has fewer connections to the sinoatrial node, compared with the right VN, and the left VNS thus induces fewer detrimental side effects on cardiac activity in both humans and rodents, and (2) the left VNS has been shown to be sufficient to induce anxiolytic effects and affect emotion-related functions. We described this rationale in the Methods part (Line 459-463).

The reasons why we selected PFC and AMY are because (1) the PFC and AMY have been widely studied in emotional research fields and suggested as the principal brain regions that express anxiety and (2) their interregional coordination of neuronal oscillations modulate anxiogenic behavior. We described this rationale in the Introduction part (Line 38-41).

As shown in Figure 2a, the left VNS induced significantly larger proportions of c-Fos-positive neurons in the PFC. Taken together with the rationales and these histological results, we mainly analyzed PFC and AMY activity patterns in this study.

8. There is no explanation for why the salivary gland was used for the reference electrode.

(Answer)

To minimize noise artifacts of VN recordings (such as myoelectric potential and minor physical fluctuation of electrode), setting its reference electrode with electrical isolation as close as possible to the recorded site is important for our VN recordings. For this purpose, the salivary gland is the best position as it is located directly above a cuff-shaped electrode implanted on the VN.

We added this explanation in the Methods part (Line 467-471).

9. There is no explanation for why the VNS parameters were selected.

(Answer)

Thank you for this crucial comment. Actually, a crucial issue in VNS both in basic and clinical studies for psychiatric and the other peripheral symptoms is that its parameters are not consistent across many studies. For example, various parameters used in previous digestive

system studies are discussed in a review paper (e.g. Table 1; Payne et al., Nat. Rev. Gastroenterol. Hepatol, 2019).

Our VNS protocol was as follows:

amplitude: 0.8 mA; pulse width: 0.1 ms; frequency: 20 Hz; duration: 30 s ON and 30 s OFF; total duration: 3 hours.

These parameters were employed based on those reported from previous studies (shown below) so that our parameters were not too strong or too weak but were within the same ranges of parameters used in the previous studies. We cited following papers and added this explanation in the Methods part (Line 511-514).

Representative papers we referred to were as follows:

- Noble et al., Brain Stimul., 2019

amplitude: 0.4 mA; pulse width: 0.1 ms; frequency: 20 Hz; duration: 30 s ON and 5 min OFF; total duration: more than days.

- Biggio et al., Int. J. Neuropsychopharmacol., 2009

amplitude: 1.5 mA; pulse width: 0.5 ms; frequency: 30 Hz; duration: 30 s ON and 5 min OFF; total duration: 3 hours to 1 months.

- Choudhary et al., Bioelectronic Medicine, 2022

amplitude: 0.2 mA; pulse width: 0.5 ms; frequency: 30 Hz; duration: 10 s ON and 30 s OFF; total duration: 2 hours.

- Furmaga et al., Biol. Psychiatry, 2011

amplitude: 0.25 mA; pulse width: 0.25 ms; frequency: 20 Hz; duration: 30 s ON and 5 min OFF; total duration: 2 weeks.

10. Testing on the elevated plus maze seems to be given over a range of 10-20 min. It is not clear if some mice explored for 10 min while others explored for 20 min, and whether they were compared across conditions. This is important because behavior on the maze changes over time.

(Answer)

We apologize this confusion of our statement. While our recording duration on the EPM test varied across mice (10-20 min), we only analyzed the first 10 min in all mice. We described it in the Methods part (Line 537-538).

11. Interesting vagus nerve spike data were acquired during the rest period, which preceded time on the elevated plus maze. It is understandable that the rest periods provided better conditions for recording, but it may be even more interesting to measure vagus nerve activity after testing on the elevated plus maze, based on research by James McGaugh and Cedric Williams indicating that

adrenaline exerts its effects on the brain AFTER exposure to stressful events, via vagus nerve signaling.

(Answer)

Thank you for this insightful comment. According to the comment, we performed recordings during rest periods after testing the EPM test (termed post-rest periods), as well as pre-rest periods presented in Figure 1f. In all mouse groups tested, we found no significant differences in both VN spike rates and VN spike power between the pre-rest and post-rest periods (presented in Supplementary Fig. 3). In this manuscript, we simply stated this result without further interpretations as follows: “basal VN activity during quiescent periods is not prominently affected by anxiety-related experiences” (Line 103-108).

As these results were observed only from a single behavioral test (EPM test), we cannot exclude the possibility that the other emotion-related experiences (e.g. more stressful events) might affect basal VN activity, as the reviewer suggested. Further studies are necessary to clarify this issue.

12. It is interesting that there are no vagus nerve activity differences between susceptible and resilient mice. Likewise, PFC and amygdala activity were not compared across susceptible and resilient mice. This makes one wonder whether exposure to social defeat, susceptibility to stress, or the combination mediates these effects.

(Answer)

Thank you for this crucial suggestion. To address the comments from the other reviewers (Reviewer 1 Comment #2 and Reviewer 3 Comment #4), we increased the number of mice, especially for resilient mice (Figure 1f (VN), 1g, 1h (behavior), 1i, 1j (VN), 3a, 3b, 3e, and 3f (PFC and AMY)). The new datasets demonstrated that resilient mice, similar to naïve mice, showed significantly larger VN spike power and higher VN spike rates than stress-susceptible mice. These results demonstrate that intrinsic VN activity during quiescent periods was reduced in susceptible mice, but not in resilient mice. These are described in Line 92-103.

As for LFP patterns, Figure 3a and 3b demonstrate that, in resilient mice, 20–30 Hz LFP power in the PFC and AMY in the open arms and during move periods in the closed arms was significantly higher than that during stop periods in the closed arms, as observed from naïve mice. Furthermore, we applied across-group comparisons (Fig. 3e) and confirmed that naïve and resilient mice showed significantly higher and lower PFC 20–30 Hz power in the open arms and during stop states in the closed arms, respectively, than stress-susceptible mice. Taken together, these results demonstrate that anxiety-related LFP

power changes in the PFC-AMY circuits were specifically disrupted in stress-susceptible mice, but not in stress-resilient mice, consistent with the results of VN power changes. These are described in Line 218-252.

13. Overall, this is a cohesive set of experiments representing work that will be of significance to the field and related fields.

(Answer)

Thank you for your evaluation of our manuscript.

Reviewer #3 (Remarks to the Author):

Overall comments:

This is an interesting study investigating the effects of stress on vagal induced oscillations in the brain

We sincerely thank the reviewer for their thorough review of our manuscript and for recognizing the significance of our work.

(1) The subdivision of animals into susceptible & resilient needs to be justified as many animals will actually be in the middle. Wouldn't extremes be better to determine stress sensitivity.

(Answer)

Thank you for this interesting suggestion. As the reviewer pointed out, the definitions of stress susceptibility in this research area are always automatic by a social interaction test. We agree that confirmations with extreme samples are important.

To exclude mice with intermediate SI ratios that were close to 1, the threshold to define stress susceptibility, we further divided each mouse group into two subgroups with higher and lower SI ratios (Supplementary Fig. 2a). A subgroup of resilience mice with higher SI ratios (>1.48 ; top 50%) was classified as strongly stress-resilient mice ($n = 10$ mice), whereas a subgroup of susceptible mice with lower SI ratios (<0.44 ; bottom 50%) was classified as strongly stress-susceptible mice ($n = 8$ mice) (Line 65-70)

We applied the same electrophysiological to these subgroups in Supplementary Figure 2b-2d and confirmed the same statistical results as those observed when all stress-resilient or stress-susceptible mice are analyzed. The main results are as follows:

- Naïve mice and strongly resilient mice showed significantly higher VN spike rates, compared with strongly susceptible mice (Supplementary Fig. 2b). (described in Line 100-102)
- Similar to stress-resilience mice, strongly resilient mice showed significantly higher VN spike power in the open arms or during move states in the closed arms than during stop states in the closed arms (Supplementary Fig. 2c, leftmost). (described in Line 143-145)
- Similar to stress-susceptible mice, such significant differences in VN spike power were not observed from strongly susceptible mice (Supplementary Fig. 2d, leftmost). (described in Line 147-149)
- Similar to stress-resilience mice, strongly resilient mice showed significantly higher PFC 20–30 Hz power in the open arms or during move states in the closed arms than during stop states in the closed arms (Supplementary Fig. 2c, right). (described in Line 230-231)
- Similar to stress-susceptible mice, such significant differences in PFC LFP patterns were not observed from strongly susceptible mice (Supplementary Fig. 2d, right). (described

in Line 241-243)

(2) The Elevated plus maze is not the best task to assess amygdala function. Fear conditioning based assays would be better.

(Answer)

Thank you for this crucial suggestion. We did have the same motivation and our preliminary strategy in this study was actually to use both fear conditioning and elevated plus maze tests. However, in our experiments with electrode implantations on the VN and brain, our electrophysiological recordings after foot shock became unstable with severe noise and loss of normal signals, possibly because of partial dysfunctions of electrodes.

In addition, electrical foot shock itself caused unexpected behavioral symptoms of mice such as strong painful behavior and fearful behavior. While exact reasons for these unexpected behavioral effects remain unknown, a possible reason is because implanted metal electrodes serve as a conductor to integrate or collect massive electrical currents from foot shock. As a result, the mice seemed to receive stronger pain than expected. Another issue was that mice implanted with electrodes seemed to be not tolerable to general levels of foot shock due to additional weights of electrodes and physical tension around the VN. Considering animal ethics, these issues were undesirable.

Due to these complex reasons, we abandoned testing fear conditioning paradigms with electrical shock in this study and only employed EPM test paradigms.

(3) The fact that stress does not alter the main parameters in the EPM is also problematic.

(Answer)

Thank you for this crucial comment. As the reviewer suggested, one of the most crucial parameters in the EPM test is “open arm time”. However, as described in the Discussion part (Line 345-347), the effects of SD stress on time in open/closed arms are not consistent among previous studies, probably due to differences in experimental conditions. To tell the truth, when we first designed this study, we also expected to see this behavioral change (open arm time). While we could not detect significant reductions in open arm time, we instead noticed that stress-susceptible mice apparently showed unstable behaviors when they entered into closed arms. We therefore defined and quantified the proportion of move states within closed arms. Interestingly, these behavioral changes were also observed from vagotomized mice and could be restored by VNS. While we cannot discuss the exact mechanisms underlying this behavior, we would like to report these consistent results as a new measure of anxiety-related behavior supported by vagus nerve activity and LFP

oscillations found in this study. These are discussed in Line 348-355.

(4) The number of animals used is small throughout and would need justification.

(Answer)

We performed additional experiments and increased the number of samples of electrophysiological recordings as follows:

(The variation in sample numbers arises from differences in the accurate placement of electrodes in each recording site)

- Naïve mice: (Figure 1 and 2): from 6–12 to 9–16 mice
- Stress-susceptible mice (Figure 1 and 3): from 5–10 to 10–15 mice
- Stress-resilient mice (Figure 1 and 3): from 0 (not shown) to 10–12 mice
- Vagotomized mice (Figure 4): from 6 to 11 mice
- Susceptible+VNS mice (Figure 5): from 5–6 to 7–11 mice

The increased sample numbers have enabled more reliable interpretations of our datasets based on statistically significant values.

All of the numbers of samples are highlighted in blue in the manuscript.

(5) The use of a single sex needs justification

(Answer)

As the reviewer indicated, only male mice were used in this study.

- In the Abstract, we added “male mice” (Line 12).
- In the Materials and Methods, we stated “male mice” at the section of “Ethical approval and animals”.
- In the Nature Portfolio Reporting summary, we described reasons why only male mice were included in this study as follows: “Only male mice were used in this study because the experimental paradigms of social defeat stress employed in this study has been established for male mice. Importantly, the stress models using male aggressor mice are only applicable to male mice.”.

REVIEWER COMMENTS

Reviewer #1 (Remarks to the Author):

The authors have done a great job in addressing my comments. I particularly appreciate the effort of the authors to add new data from resilient animals, which strengthens the conclusion of the study.

Since only male mice are used in the study, the title of the paper should be modified to show that findings from this study were obtained from male mice only.

Tak Pan Wong
Douglas Research Centre
McGill University

Reviewer #2 (Remarks to the Author):

This resubmission was responsive to reviewer comments; however, I still have a few concerns.

Reviewer 3 recommended using only mice scoring on the extreme ends in the SI test, rather than using the less than or greater than 1 criterion. The suggestion makes good sense because animals scoring in the middle could dilute a real predisposition effect. Unfortunately, the response was to add the data for extreme subdivisions: "strongly stress-resilient" and "strongly stress-susceptible" for some of the tests, but not all. Statistics that are reported for analysis of the subgroup data do not include F or P values and data are shown in the supplementary materials. Presumably, the authors report only those tests that show results consistent with those seen with the larger groups. For example, testing time spent in the open arm for strongly stress-susceptible vs. strongly stress-resilient mice would be a preferable way to test the hypothesis that susceptible mice are more anxious than resilient mice, but that analysis is not included. Given this selectivity, it is not clear that using the extreme groups adds value. In reporting results, the back-and-forth between the larger groups and extreme subgroups is likely to frustrate readers. It might work better as a secondary analysis, that is described with the rationale and includes statistical results, in a separate paragraph or section of the paper.

On page 5, line 69-70, the N for each group appears to be mixed up. According to the figure (S2), N=8 resilient and N=10 susceptible.

Only 12 resilient and 12 susceptible mice were used in the VN recording experiment. This is not the number identified by (less than or greater than 1) criterion or the number included in the extreme subgroups. What was the process for selecting the mice for this study?

Although two experts in the revision process proofread the manuscript, there are several areas where the word "respectively" is used in a way that confuses the meaning of the sentence so much that it may mislead readers to misunderstand the reported results (page 2, line 13-15; page 4, line 58-60; page 9, line 181-183, for example). In addition, "vagotomography" should be changed to "vagotomy" on page 3, line 29.

The claim that increased locomotion in the closed arms is a measure of anxiety was addressed with a reference and some softening of the language. The reference is a single paper arguing that changes in cortical oscillations are influenced by activity in the closed arms. This reasoning is a bit circular, but it would be appropriate to make the point that moving state "has been used as a measure of stress-induced anxiety", rather than "is considered a behavioral measure of stress-induced anxiety" (page 17 line 350-351), and an acknowledgement that these findings are also consistent with alternative explanations such as a relationship between vagal activity and exploratory drive or hyperactivity,

although those hypotheses are not as well supported by the literature as the anxiety hypothesis.

The new analysis showing that the SI scores are stable for 3 weeks is encouraging. I recommend adding the second test to the timeline in Figure 1. It is also encouraging that there are now statistically significant differences between resilient and susceptible animals in VN activity and LFP patterns.

Once again, the findings are interesting. Additional data collection has improved this revised version of the manuscript, but careful revisions of the writing will make the work more accessible and compelling for readers.

Reviewer #3 (Remarks to the Author):

My comments have been addressed.

Despite the residual issues which are well reflected on the paper will be an interesting addition to the literature

Manuscript number: NCOMMS-23-03900A

"Stress-induced vagal activity influences anxiety-relevant prefrontal and amygdala neuronal oscillations in male mice"

Reviewer #1 (Tak Pan Wong):

The authors have done a great job in addressing my comments. I particularly appreciate the effort of the authors to add new data from resilient animals, which strengthens the conclusion of the study.

We wholeheartedly appreciate his recognition of the significance of our work.

Since only male mice are used in the study, the title of the paper should be modified to show that findings from this study were obtained from male mice only.

We corrected our title:

"Stress-induced vagal activity influences anxiety-relevant prefrontal and amygdala neuronal oscillations in male mice"

Reviewer #2:

(1) Reviewer 3 recommended using only mice scoring on the extreme ends in the SI test, rather than using the less than or greater than 1 criterion. The suggestion makes good sense because animals scoring in the middle could dilute a real predisposition effect. Unfortunately, the response was to add the data for extreme subdivisions: "strongly stress-resilient" and "strongly stress-susceptible" for some of the tests, but not all.

Statistics that are reported for analysis of the subgroup data do not include F or P values and data are shown in the supplementary materials. Presumably, the authors report only those tests that show results consistent with those seen with the larger groups.

For example, testing time spent in the open arm for strongly stress-susceptible vs. strongly stress-resilient mice would be a preferable way to test the hypothesis that susceptible mice are more anxious than resilient mice, but that analysis is not included. Given this selectivity, it is not clear that using the extreme groups adds value.

(Answer)

Thank you for this valuable comment. As the reviewer pointed out, the aim of our analysis in the previous manuscript was to report that the results from extreme groups are consistent with those seen with the larger groups.

According to the comment, we now analyzed whether there were more pronounced behavioral changes from the extreme groups (strongly susceptible and strongly resilient groups) using the across-group comparisons in newly added Supplementary Figure 2c. After all, we found the similar behavioral patterns from these extreme mouse groups to those observed from all the susceptible and resilient mice (Fig. 1h and 1i). These results confirm that, at least from our behavioral experiments, no more pronounced behavioral differences were observed even when we focused on the extreme groups. These behavioral results are described in the Results part (Line 162-171).

In reporting results, the back-and-forth between the larger groups and extreme subgroups is likely to frustrate readers. It might work better as a secondary analysis, that is described with the rationale and includes statistical results, in a separate paragraph or section of the paper.

(Answer)

Thank you for this constructive comment. We now created an independent paragraph that summarizes all behavioral tests and VN spike recordings including the relevant statistical results (that were presented in the Supplementary Legends in the previous version) for the strongly resilient and susceptible groups (Line 151-182).

As for LFP results, we have included the relevant statistical results in the appropriate sections (Line 256-261 and Line 271-277).

(2) On page 5, line 69-70, the N for each group appears to be mixed up. According to the figure (S2), N=8 resilient and N=10 susceptible.

(Answer)

Thank you for finding these errors. We now corrected the numbers (Line 156-157).

(3) Only 12 resilient and 12 susceptible mice were used in the VN recording experiment. This is not the number identified by (less than or greater than 1) criterion or the number included in the extreme

subgroups. What was the process for selecting the mice for this study?

(Answer)

This was because VN recordings were not successful from some of the mice identified in behavioral tests.

In all experiments, after behavioral assessments, we first tried to obtain simultaneous electrophysiological recordings (VN activity, EMG signals, and PFC and AMY LFP signals), but some electrodes were implanted outside the targets or accidentally disrupted during chronic implantations. Even from these mice, we recorded as many signals as possible simultaneously. We now explained these recording conditions in the Methods part (Line 519-521 and Line 528-532).

All the actual numbers (n) of samples used for analyses are described throughout the manuscript.

(4) Although two experts in the revision process proofread the manuscript, there are several areas where the word "respectively" is used in a way that confuses the meaning of the sentence so much that it may mislead readers to misunderstand the reported results (page 2, line 13-15; page 4, line 58-60; page 9, line 181-183, for example).

In addition, "vagotomography" should be changed to "vagotomy" on page 3, line 29.

(Answer)

We apologize for these errors after the proofreading. We have made corrections on the following points.

Corrected to "instantaneous spike rates of the vagus nerve were negatively and positively correlated with the power of 2–4 Hz and 20–30 Hz oscillations, respectively, in the prefrontal cortex and amygdala" (Line 13-15)

Corrected to "Of the 36 mice that received SD stress, 20 mice had SI ratios less than 1 and were classified as stress-susceptible mice, whereas 16 mice had SI ratios more than 1 and were classified as stress-resilient mice (Fig. 1b)" (Line 58-60)

Corrected to "The same significantly negative correlations were observed from AMY 2–4 Hz LFP signals in 4 mice out of 9 naïve mice tested, whereas significantly positive correlations were observed from AMY 20–30 Hz LFP signals in all the 9 mice tested (Fig. 2d, bottom)." (Line 206-209)

Corrected to "vagotomy" (Line 29).

(5) The claim that increased locomotion in the closed arms is a measure of anxiety was addressed with a reference and some softening of the language. The reference is a single paper arguing that changes in cortical oscillations are influenced by activity in the closed arms. This reasoning is a bit circular, but it would be appropriate to make the point that moving state "has been used as a measure of stress-induced anxiety", rather than "is considered a behavioral measure of stress-induced anxiety" (page 17 line 350-351), and an acknowledgement that these findings are also consistent with alternative explanations such as a relationship between vagal activity and exploratory drive or hyperactivity, although those hypotheses are not as well supported by the literature as the anxiety hypothesis.

(Answer)

Thank you for these very meaningful suggestions. We agree with the reviewer that our behavioral result (increased locomotion) could be interpreted as a measure of anxiety. We rewrote the Discussion part as suggested: "we detected a difference between resilient and susceptible phenotypes when focusing on the proportion of move states within closed arms, which has been used as a measure of anxiety (the reference)" (Line 383-385)

As for the hypothesis of the relationship between vagal activity and exploratory drive or hyperactivity (as suggested in the last sentence), this is a very interesting idea and we have considered once again whether to describe this possibility. However, due to the lack of substantial literature supporting this relationship, as concerned by the reviewer, we have decided not to further mention it here in this paper.

(6) The new analysis showing that the SI scores are stable for 3 weeks is encouraging. I recommend adding the second test to the timeline in Figure 1. It is also encouraging that there are now statistically significant differences between resilient and susceptible animals in VN activity and LFP patterns.

(Answer)

Thank you for this constructive comment. We moved the previous Supplementary Figure 1 to the main Figure 1c (the legends were moved to the main manuscript) (Line 60-68). Thank you for your encouraging comment on the LFP patterns.

(7) Once again, the findings are interesting. Additional data collection has improved this revised version of the manuscript, but careful revisions of the writing will make the work more accessible and compelling for readers.

We would like to express our sincere gratitude to the reviewer for acknowledging the significance of our work. Thanks to the insightful comments, we are confident that the quality of this manuscript has been significantly improved.

Reviewer #3 (Remarks to the Author):

My comments have been addressed.

Despite the residual issues which are well reflected on the paper will be an interesting addition to the literature.

We wholeheartedly appreciate her/his recognition of the significance of our work.

REVIEWERS' COMMENTS

Reviewer #2 (Remarks to the Author):

My comments were addressed. This will make a nice contribution to the field.

Manuscript number: NCOMMS-23-03900B

"Stress-induced vagal activity influences anxiety-relevant prefrontal and amygdala neuronal oscillations in male mice"

Reviewer #2:

My comments were addressed. This will make a nice contribution to the field.

(Answer) We wholeheartedly appreciate her/his recognition of the significance of our work.